# Geographically dispersed zoonotic tuberculosis in pre-contact South American human populations

Åshild J. Vågene [1,2,3,14 ✉], Tanvi P. Honap [4,5,6,14 ✉], Kelly M. Harkins[7], Michael S. Rosenberg [4,8], Karen Giffin[1,9], Felipe Cárdenas-Arroyo[10], Laura Paloma Leguizamón [10], Judith Arnett [7,11], Jane E. Buikstra [7], Alexander Herbig [1,2,9], Johannes Krause [1,2,9,15 ✉], Anne C. Stone [7,12,13,15 ✉] & Kirsten I. Bos [1,2,9,15 ✉]

Previous ancient DNA research has shown that *Mycobacterium pinnipedii*, which today causes tuberculosis (TB) primarily in pinnipeds, infected human populations living in the coastal areas of Peru prior to European colonization. Skeletal evidence indicates the presence of TB in several pre-colonial South and North American populations with minimal access to marine resources— a scenario incompatible with TB transmission directly from infected pinnipeds or their tissues. In this study, we investigate the causative agent of TB in ten pre-colonial, non-coastal individuals from South America. We reconstruct *M. pinnipedii* genomes (10- to 15-fold mean coverage) from three contemporaneous individuals from inland Peru and Colombia, demonstrating the widespread dissemination of *M. pinnipedii* beyond the coast, either through human-to-human and/or animal-mediated routes. Overall, our study suggests that TB transmission in the pre-colonial era Americas involved a more complex transmission pathway than simple pinniped-to-human transfer.

[1] Department of Archaeogenetics, Max Planck Institute for the Science of Human History, Jena, Germany. [2] Institute for Archaeological Sciences, University of Tübingen, Tübingen, Germany. [3] Section for Evolutionary Genomics, GLOBE Institute, University of Copenhagen, Copenhagen, Denmark. [4] School of Life Sciences, Arizona State University, Tempe, AZ, USA. [5] Department of Anthropology, University of Oklahoma, Norman, OK, USA. [6] Laboratories of Molecular Anthropology and Microbiome Research, University of Oklahoma, Norman, OK, USA. [7] School of Human Evolution and Social Change, Arizona State University, Tempe, AZ, USA. [8] Center for Biological Data Science, Virginia Commonwealth University, Richmond, VA, USA. [9] Department of Archaeogenetics, Max Planck Institute for Evolutionary Anthropology, Leipzig, Germany. [10] Colombian Institute of Anthropology and History (ICANH), Bogotá, Colombia. [11] University of the Andes, School of Medicine, Bogotá, Colombia. [12] Center for Evolution and Medicine, Arizona State University, Tempe, AZ, USA. [13] Institute of Human Origins, Arizona State University, Tempe, AZ, USA. [14] These authors contributed equally: Åshild J. Vågene, Tanvi P. Honap. [15] These authors jointly supervised: Johannes Krause, Anne C. Stone, Kirsten I. Bos. ✉email: ashild.vagene@sund.ku.dk; thonap@asu.edu; krause@eva.mpg.de; acstone@asu.edu; kirsten_bos@eva.mpg.de

The majority of modern human tuberculosis (TB) infections are caused by *Mycobacterium tuberculosis sensu stricto* and *M. africanum* strains that comprise nine human-adapted lineages (referred to as Lineages 1-9)[1–3]. These lineages, together with several animal-associated strains and the ancestral smooth tubercle bacilli, *M. canettii*, form the *Mycobacterium tuberculosis* complex (MTBC).

Today, the number of reported human TB cases caused by animal-associated strains is low. In 2019, for example, the WHO estimated only 140,000 TB cases of a zoonotic origin[4]; however, this is likely an underrepresentation since clinical strain assignment is rarely performed in developing nations[5]. The majority of zoonotic TB cases in humans are attributed to *M. bovis*, which predominantly causes TB infection in cattle, and to a lesser extent *M. caprae*, which is associated with domestic sheep and goats[6]. The notion that MTBC is comprised of host-specific strains[7] is continually challenged by examples that document inter-species transfer between wild, captive, and domesticated animals, and humans[5,6,8–12]. Our limited knowledge about the natural host ranges and zoonotic capacities of these strains makes them an underappreciated threat to both human and animal health in many parts of the world[8,13].

Genomic data from ancient bacterial and viral organisms have revealed important details regarding disease ecology in the past[14,15]. In 2014, Bos, et al.[12] reported the complete genome sequences for three MTBC strains recovered from archaeological skeletal remains of human individuals from the southern coast of Peru, which predate European contact in the Americas. Skeletal pathology has indicated the presence of TB in indigenous peoples of the Americas long before European contact[16,17], an observation that was difficult to reconcile with the dominance of modern European *M. tuberculosis* lineages in the Americas today[18]. These ancient Peruvian strains are distinct from human-adapted MTBC and are genetically most closely related to modern *M. pinnipedii*. The *M. pinnipedii* lineage is associated with pinnipeds (seals and sea lions) and rarely infects humans today[19–21]. Ancient zoonotic events related to the manipulation and consumption of infectious seal tissues are hypothesized to account for the transmission of *M. pinnipedii* strains to pre-contact humans who occupied this coastal region[12].

The archaeological observations of human skeletal and desiccated soft tissue remains displaying pathological changes consistent with TB infection have been recorded in many pre-contact archaeological sites across South and North America[16,17]. The earliest cases are found in Peru and northern Chile and are dated to ~700 CE, with possible cases occurring as early as 290 CE[16,22,23]. Peru and northern Chile also have the highest density of archaeological TB cases in South America. By contrast, affected human remains from northern regions of South America are fewer in number. Cases of pre-contact TB in North America begin to appear in the archaeological record after 900 CE, and the majority of affected individuals are located at inland sites in the midcontinent and southwestern USA[16]. The presence of putative TB cases at inland sites across the Americas, in places where direct contact between pinniped and human populations would have been unlikely, invites a more thorough sampling of ancient human remains to better understand past genetic diversity and the temporal and geographic distribution of MTBC strains circulating in the pre-contact Americas.

Here, we present three additional ancient South American MTBC genomes isolated from pre-contact human skeletal remains. All three strains belong to the *M. pinnipedii* clade and thus reveal its widespread geographical presence. Importantly, these new ancient *M. pinnipedii* genomes were recovered from individuals from inland Peruvian and Colombian sites where human contact with infectious pinniped tissues is unlikely to have occurred. This finding reveals a more complex transmission pathway that moves beyond a simple pinniped-to-human zoonotic event.

## Results

**Screening**. A total of ten individuals showing signs of skeletal TB were screened for this study. DNA extracts made from bone powder from four individuals (82, 281, 382, and 386) were considered positive for MTBC DNA based on qPCR assay results. Our extraction negative controls did not yield detectable amplification products (Supplementary Data 1). Bone from individual 281 was extracted three times over the course of this study: these extracts are referred to as 281a, 281b, and 281c. Positive qPCR results for extracts 82 and 281a have already been published elsewhere[24]. Second-tier screening via gene capture of non-uracil-DNA-glycosylase treated libraries (non-UDG; nU), targeting five mycobacterial genes, *rpoB*, *gyrA*, *gyrB*, *katG*, and the mtp40 segment of the *plcA* gene, yielded positive results for 281bnU and 386nU wherein all five genes were covered. The MTBC gene capture results for 82nU were also previously published[12]. All extracts from samples 82, 281, 386 were selected for in-solution whole-genome capture of the MTBC genome. Library 382nU yielded negligible coverage of *katG* and mtp40 segment, which are specific to the MTBC, and therefore, was not included in the whole-genome capture. The extract for 281c was not tested via qPCR or gene capture, but it was included in the whole genome in-solution capture based on the positive screening results for 281anU and 281bnU. Screening results are summarized in Supplementary Data 1.

**In-solution genome capture**. DNA extracts 82, 281a, 281b, 281c, and 386, along with their associated negative controls, were processed into non-UDG (nU) and UDG-treated (U) libraries, each differentiated by a unique set of molecular indices. After in-solution MTBC genome capture, the UDG-treated libraries showed endogenous DNA content ranging from 0.71-1.66% from reference-based mapping (Supplementary Data 2). Samples 82, 281a, 281b, 281c, and 386 experienced respective enrichments of 18-, 42-, 43-, 31-, and 43-fold. A positive control library (sample 58U), generated in our earlier study[12], had a 64-fold enrichment with 41.67% endogenous DNA after capture (Supplementary Data 2), indicating that the low percent endogenous DNA observed in the other libraries was not caused by experimental conditions. Metagenomic analyses of shotgun and capture data for the UDG-treated libraries were conducted using the Megan ALignment Tool (MALT)[25]. For the UDG-treated shotgun data, only 0.004–0.012% of all reads were assigned to members of the MTBC at an 85% minimum identity threshold. 9.7–14.5% of reads were assigned to other taxa, including 0.015–0.072% to non-MTBC mycobacterial species (Supplementary Data 3 and 4). The remaining reads from our sequencing data could not be assigned based on our taxonomic reference set, which consisted of the full NCBI Nucleotide (nt) database (7th December 2016). MALT analysis of the capture data for the UDG-treated libraries showed that these non-target reads persisted after MTBC-genome enrichment, where only 0.3–1% of all reads were assigned to the MTBC, and 17.4–24.7% of reads were assigned to other taxa including 0.09–0.34% to non-MTBC mycobacterial species (Supplementary Data 3 and 4). This demonstrates that our MTBC capture assay enriches DNA from genetically similar mycobacterial species in addition to targeted MTBC strains. The UDG-treated captured library for 281cU was not sequenced deeper since it had lower levels of MALT-assigned endogenous MTBC reads (0.337%) as compared to 281aU (1.013%) and 281bU (0.7%) after capture (Supplementary Data 4). The proportion of MALT-assigned MTBC versus non-MTBC reads is further illustrated in Supplementary Figure 1.

**Table 1 MTBC genome mapping statistics for UDG-treated enriched libraries.**

| Sample ID | # Processed reads before mapping | # Unique quality-filtered mapped reads | Endogenous DNA (%) | Mean fold coverage | % of genome covered at least fivefold | Median fragment length (bp) |
|---|---|---|---|---|---|---|
| 82U | 449,538,686 | 1,002,140 | 0.71 | 14.95 | 80.31 | 75 |
| 281U | 170,865,535 | 1,003,454 | 1.30 | 15.34 | 80.81 | 76 |
| 386U | 328,622,471 | 736,677 | 1.66 | 10.80 | 83.07 | 71 |

**Analysis of non-MTBC mycobacterial contamination.** We compared the ratios of MTBC and non-MTBC mycobacterial reads present in three of our sample libraries (82nU, 281bnU, 386nU) to that of the previously published Peruvian samples[12]. This procedure assessed the level of contaminant mycobacterial reads present in our samples and allowed us to anticipate whether such closely related bacteria could interfere with downstream analyses. Using the database described above, MALT results for the non-UDG treated shotgun data were generated using both 85 and 95% minimum-percent-identity thresholds to calculate the ratio of MTBC to non-MTBC mycobacterial reads (Supplementary Data 5 and 6). The previously published Peruvian samples (54nU, 58nU, and 64nU) contain 0.022–0.242 non-MTBC mycobacterial reads for every MTBC assigned read at 85% identity, and 0.013–0.046 at 95% identity (Supplementary Data 7). The ratio of non-MTBC mycobacterial DNA in the samples from this study is remarkably higher with 74, 5.1, and 9.2 non-MTBC mycobacterial reads for every MTBC assigned read at 85% identity, and 16, 1.1, and 1.8 reads at 95% identity for 82nU, 281bnU, and 386nU respectively (Supplementary Data 7). This demonstrates that the samples presented here contain a substantial amount of non-MTBC mycobacterial DNA background in comparison to the samples published by Bos et al.[12], and that many of these non-target reads persist after stringent (95% identity) filtering.

**Mapping statistics.** Based on the UDG-treated MTBC genome-capture data, we reconstructed complete genomes for three ancient MTBC strains, each from a separate individual. The 281aU and 281bU genomes, retrieved from a vertebra and a rib, respectively, from the same individual, were determined to be identical and the data were therefore combined (Methods; Supplementary Figure 2). From this point onwards, we refer to this composite genome as 281U. The average coverage for 82U, 281U, and 386U was 14.9-, 15.3-, and 10.8-fold, respectively, with 80-83% of the ancestral MTBC reference covered at least five-fold (Table 1, Supplementary Data 2). Reads in our extraction and library negative controls had high duplication rates after capture, indicating that the molecular content of the libraries had been sufficiently explored (Supplementary Data 2). Reference-based mapping analyses using sensitive mapping parameters revealed as many as 2,868 unique mapping reads in our negative controls, though MALT assigned few of these (between 0 to 371, with a mean of 34.7) to the MTBC and lower taxonomic nodes (Supplementary Data 2 and 8). Although the assignment of MTBC reads in non-pathological samples is a known phenomenon[26], these data indicate that the reagents were not the main source of the mycobacterial contaminants observed in our samples. None of the reference-mapped negative controls showed a damage pattern characteristic of ancient DNA.

**Authentication of ancient DNA damage patterns.** Capture data from the non-UDG treated libraries were used to determine deamination patterns for the enriched ancient MTBC DNA and co-enriched ancient human DNA. Ancient DNA damage occurs at the ends of degraded DNA fragments in the form of cytosine deamination that accumulates over time[27]. The deamination patterns for the first base at the 5′ ends for libraries 82nU, 281nU, and 386nU were initially estimated to be 4.7, 8.58, and 8.1%, respectively, for all reads mapping to the ancestral MTBC reference (Supplementary Data 2). On the assumption that low damage resulted from non-target mapping from modern organisms, we applied a selective mapping approach to filter out non-target reads. Reads that contained a high number of mismatches outside of the four bases at either ends of a read, where mismatches due to deamination are most likely to occur, were removed through the application of stricter bwa-aln mapping parameters (see Methods). Damage patterns were subsequently re-calculated from the filtered non-UDG dataset (Supplementary Data 9). Reassessment of the damage patterns yielded 7.7% (82nU), 8.7% (281nU), and 7.3% (386nU) for the first base of the 5' end of the reads, thus confirming the presence of ancient DNA[27] (Supplementary Data 9). Higher levels of deamination were observed for the human DNA (~14% for all samples) than for MTBC DNA (Supplementary Data 10). Deamination rates of ancient MTBC DNA ranging from 3–7% coupled with higher rates of deamination for human host DNA has also been recorded from contemporaneous individuals (1000–1400 CE) from Huari, Peru[28], though these assessments were made in the absence of non-target MTBC filtering, such as those performed here. The low rate of DNA deamination in ancient MTBC reads could be due to the thick mycobacterial cell wall, whereby mycolic acid decreases permeability and potentially protects the DNA from hydrolytic and enzymatic degradation over time[29], as has previously been suggested for ancient *M. leprae*[30]. However, the persistence of non-target modern DNA sequences in our dataset, despite our filtering attempts, cannot be ruled out.

**Archaeological context.** Two strains, 281U and 386U, were reconstructed from individuals from the Colombian sites of Las Delicias and Candelaria La Nueva, respectively, located in the modern city of Bogotá, which is situated more than 600 km inland (by air) and ~2640 m above sea level in the Eastern Cordillera of the Andes (Fig. 1). The third strain, 82U, was recovered from an individual from the Moquegua, M6-Estuquiña site, located at the juncture of the middle and upper Osmore valleys, ~65 km inland (by air) from the southern coast of Peru, situated about 1500 m above sea level in the Andes[31–33] (Fig. 1). Radiocarbon dates indicate that individual 281 (1265–1380 CE) pre-dates European contact, while individual 386 (1450–1640 CE) overlaps with European presence in the Bogotá region, beginning 1536–1537 C.E.[34,35] (Supplementary Data 11). The site from which individual 82 was excavated has been archaeologically dated to the pre-contact period (1250–1470 CE)[36]. Further archaeological context is provided in Supplementary Note 1.

**Phylogenetic analyses and molecular dating of ancient MTBC.** A dataset comprising our three new ancient MTBC genomes, 261 ancient and modern genomes from the dataset used by

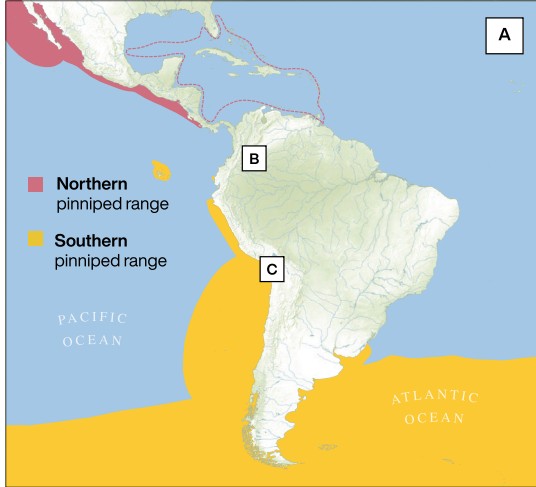

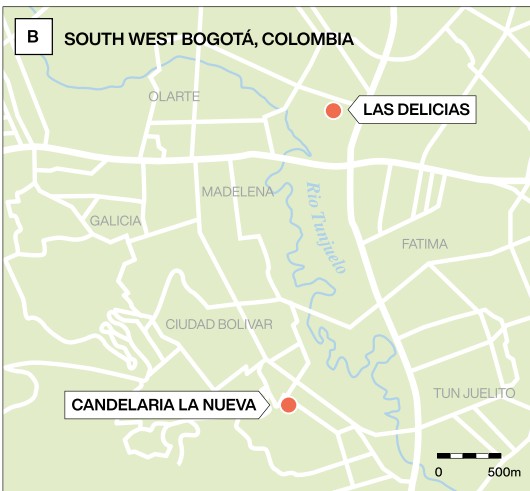

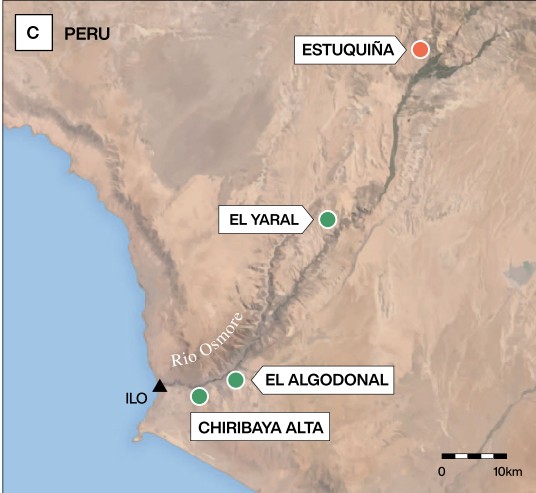

- ● Archaeological site, this study
- ● Archaeological site, Bos et al. 2014
- ▲ Modern city

**Fig. 1 Maps indicating the modern pinniped range and locations of archaeological sites that have yielded ancient *M. pinnipedii* genomes. A** shows a map of South and Central America illustrating the modern and historical southern (yellow) and northern (red) pinniped range. The modern ranges of *Arctocephalus australis*[102], *Arctocephalus galapagoensis*[103], *Arctocephalus philippii*[104], *Mirounga leonina*[105], *Otaria byronia*[106] and *Zalophus wollebaeki*[107] are overlaid for the southern hemisphere and *Zalophus californianus*[108] is shown for the northern hemisphere. Modern range data is from The International Union for Conservation of Nature's Red List of Threatened Species (http://www.iucnredlist.org/). The historical northern range of the Caribbean monk seal, *Monachus tropicalis*, is indicated by the dashed red line after ref. [109]. The geographical locations of panels B and C are shown; **B**) shows the locations for the Colombian sites (magnified); **C** shows the Lower and Middle Osmore River Valley in Peru (magnified). The locations of the sites that yielded *M. pinnipedii* genomes in this study and the published study by Bos, et al.[12] are shown. Graphic produced by Michelle O'Reilly.

and form a clade basal to the ancient Peruvian and modern *M. pinnipedii* strains. The new ancient Peruvian strain, 82U, clusters with those previously published from the region and modern *M. pinnipedii* strains but diverges earlier.

All six ancient *M. pinnipedii* genomes show some degree of branch shortening– such a trend is expected of ancient data since less time has passed for mutations to accumulate. Despite all ancient strains being almost contemporaneous, longer branch lengths were observed for the new genomes (82U, 281U, and 386U). This could be interpreted as a higher number of derived positions in the new data as compared to the previously published genomes (54U, 58U, and 64U)[12] (Supplementary Fig. 3). An investigation of multiallelic variant calls showed that 82U, 281U, and 386U have higher numbers of multiallelic sites when compared to the three previously published Peruvian genomes (Supplementary Fig. 4). We believe this is best explained by the presence of genetically similar non-target DNA in the bones stemming from environmental taxa that were coenriched during capture. These reads remain in our mapped dataset despite our adherence to stringent mapping parameters. A dominance of genetically similar non-target reads in certain loci could cause SNPs to be erroneously called, despite the application of our 90% homozygosity threshold.

On the suspicion that non-target reads erroneously inflated the number of homozygous SNPs in genomes 82U, 281U, and 386U, we investigated reads covering all homozygous SNPs unique to each of these three genomes, as well as those shared only between 281U and 386U. Using a BLASTN[39] search against the NCBI-nt database (April, 2021), we removed reads where at least one non-MTBC species was listed amongst the top five matches (Methods). Our tree construction, based on complete deletion, revealed this process to have only marginal influence on the constructed tree, as only one derived homozygous SNP from genome 82U was removed, and this had no impact on the topology (Fig. 2, Supplementary Figs. 3 and 5).

We used BEAST v1.8[40] to conduct Bayesian-based molecular dating analysis using a SNP alignment of all *M. pinnipedii* strains (ancient and modern) and *M. microti*. The radiocarbon/archaeological dates as calibration points. We estimate that all *M. pinnipedii* strains shared a common ancestor about 1408 years before present (YBP) with a 95% Highest Posterior Density (HPD) interval of 1077–1819 YBP (Supplementary Fig. 6), which is congruent with previous estimates[12]. However, we observe wider 95% HPD intervals in this analysis compared to the earlier study[12], which could be potentially explained by the persistence of non-target reads in our data.

Bos, et al.[12] and two additional animal-associated strains[37,38], was used for phylogenetic analyses (Supplementary Data 12). Our new ancient genomes were positioned with the previously published ancient Peruvian and modern *M. pinnipedii* genomes and retained the same phylogenetic positioning regardless of the tree-building method (Fig. 2; Supplementary Figs. 3 and 4). The ancient Colombian strains (281U and 386U) are closely related

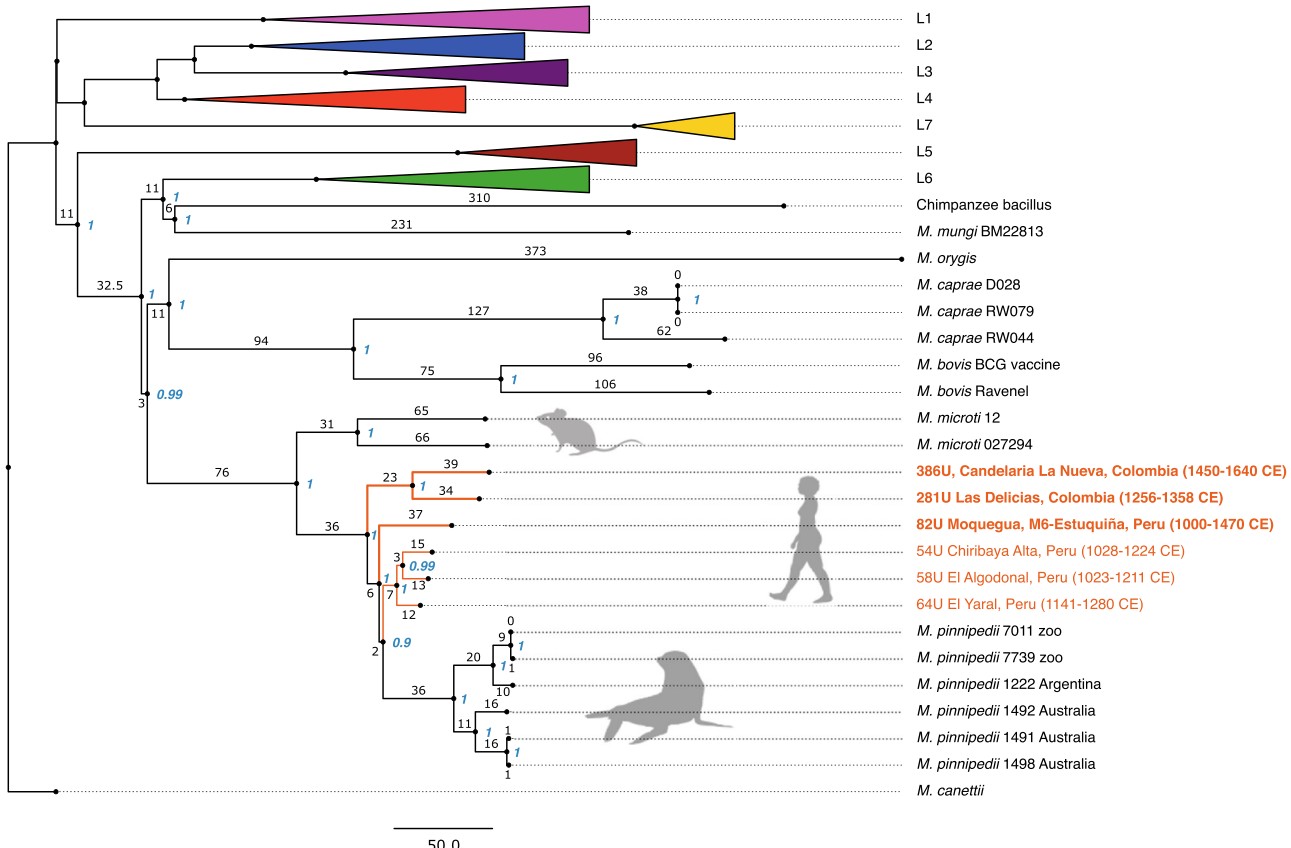

**Fig. 2 Maximum Parsimony MTBC phylogeny.** The tree was constructed using the full dataset of 266 MTBC genomes, including the six ancient genomes that are highlighted in orange. Genomes 82U, 281U, and 386U are marked in bold and BAM files filtered for spurious reads were used for SNP calling before tree construction. The tree is based on 14,262 positions out of a possible 44,235, with all missing and ambiguous sites excluded, using 500 bootstrap replicates. Bootstrap support (blue) and branch lengths are marked. Human-adapted lineages 1–5 and 7, and human-associated L6 strains have been collapsed. Our three ancient genomes fall together with other ancient Peruvian genomes within the *M. pinnipedii* clade.

**SNP analysis of protein-coding genes**. After read filtering of unique and shared positions among our ancient genomes (described above; Methods), we identified 1,123 variant positions that occur in coding regions in at least one of the six ancient *M. pinnipedii* genomes. 691 SNPs create nonsynonymous amino acid changes; 30 of these create pseudogenes through the loss of two start-codons (START_LOST) and the gain of 28 stop-codons (STOP_GAINED). An additional 432 SNPs are synonymous changes (Supplementary Data 13). We did not further investigate variant calls occurring in any of our new genomes due to the inferred high levels of contamination in our data that could contribute to false-positive SNP calls. However, under our analytical parameters, a SNP previously called in Rv3768 (position 4,214,338) was determined not to be unique to the three ancient Peruvian genomes published by Bos et al.[12] (54U, 58U, and 64U), and was in fact shared by all modern and ancient genomes in the *M. pinnipedii* clade. By contrast, an additional nonsynonymous SNP occurring in gene *accD3* was found to be unique to these three Peruvian genomes (54U, 58U, and 64U). *accD3* encodes acetyl-CoA carboxylase carboxyl transferase subunit beta, which plays a key role in the biosynthesis of mycolic acid[41]. This revised SNP profile results in the same phylogenetic structure as reported previously[12].

**Regions of difference**. The presence of regions of difference (deletions) characteristic of specific MTBC animal-associated strains was investigated in our three new ancient genomes. Consistent with previous findings in ancient and modern *M.* *pinnipedii* strains[12,42], our genomes had the characteristic RD2seal deletion, specific for *M. pinnipedii*, as well as the RD7, RD8, RD9 and RD10 deletions. These deletions are evidenced by reads bridging the deletion breakpoints in the MTBC_anc reference (Methods; Supplementary Data 14). Contaminant reads in our dataset are stacking in specific regions within the deletions, but these reads contain a high number of mismatches and likely arise from non-MTBC mycobacteria. The RDmic deletion (specific to *M. microti*) was not present in our strains.

## Discussion

Our previous study provided evidence that pre-contact era TB in Peru, which has been extensively documented by bioarcheological evidence[16,17,22], was, at least in part, caused by MTBC strains belonging to the *M. pinnipedii* clade[12] (Fig. 1). These ancient *M. pinnipedii* genomes were recovered from human remains from coastal sites located in the Osmore River valley in Peru, where zoonotic pinniped-to-human transmission is hypothesized to have played a role in the bacterium's spread to these coastal human populations[12]. In our current study, we sampled human remains from inland sites in Peru and Colombia to assess whether *M. pinnipedii* accounted for TB cases observed in these regions. Our finding of ancient *M. pinnipedii* strains from individuals unlikely to have had direct contact with infected pinnipeds shows that, in antiquity, *M. pinnipedii* had a geographic range beyond the Peruvian coast. This suggests additional modes of TB transmission in these populations, such as human-to-human or terrestrial animal-to-human.

The Osmore River valley in Peru extends from the Andean altiplano to the coast (Fig. 1). Our previous study sampled individuals from the coastal Osmore valley, belonging to the Chiribaya cultures associated with the Middle Horizon/Late Intermediate period (750-1350 C.E.)[12]. The peoples inhabiting coastal Peru and northern Chile during the first millennium C.E. are known to have exploited seal tissues for sustenance and tool manufacture[43–46]. Consumption of, or close contact with, infectious seal tissues could have provided multiple opportunities for zoonotic infection to occur. Our new ancient *M. pinnipedii* strain from Moquegua, M6-Estuquiña (82U) is from the upper middle Osmore River Valley (also known as the Moquegua valley) located about 65 km inland (by air) and about 1500 m above sea level and is separated from the coast by a mountain barrier[31–33]. This site was occupied during the Late Intermediate Period (LIP; 1000-1470 C.E.) and is considered the type site for the Estuquiña culture[32,47]. Skeletal lesions consistent with prolonged TB infection are common in the M6-Estuquiña skeletal assemblage, where they occur in ~9% of the general population, and specifically in 19.2% of adult males and 9.8% in adult females[48], thus suggesting chronic TB may have been endemic in this population. The higher percentage of males with skeletal TB lesions could indicate that men were more frequently exposed to MTBC bacteria, or that they were more likely to sustain a prolonged infection allowing the formation of skeletal lesions.

Our new ancient Colombian *M. pinnipedii* strains are from the sites of Las Delicias and Candelaria La Nueva near Bogotá, which are more than 600 km inland (by air) and ~2640 m above sea level. The individuals from whom these genomes derive, belonged to the Muisca confederation, a societal structure of local tribes inhabiting the region from 950 to 1550 CE[35,49]. All known individuals with skeletal lesions indicative of TB ($n = 9$) from the Bogotá region (at the time of sampling) were tested as part of this study (Supplementary Data 1; Supplementary Note 1). Archaeological evidence suggestive of pinniped tissue exploitation is lacking from these highland inland regions[50–52]. Furthermore, stable isotope data (Supplementary Data 11), primarily δ15N values ranging from +9.9 to +10.2‰, suggest the lack of a marine vertebrate component in the diet of these individuals[53–55].

Our finding of *M. pinnipedii* infection in individuals from inland South American sites suggests a mode of transmission for this pathogen that went beyond simple pinniped-to-human zoonotic transmission. The possibility of *M. pinnipedii* having adapted to and spread among humans, after an initial zoonotic transfer, could account for the patterns noted in our data. Both the Peruvian and Colombian sites from which our ancient *M. pinnipedii* genomes derive are known to have been part of larger trade networks. The strategic location of the Moquegua, M6-Estuquiña site facilitated access to corridors leading to the upper valleys and to the coast where trade caravans would pass by, linking these populations during the LIP[32]. Remains of marine fish[32,56,57], along with artefacts and architecture[32,33,58] found at M6-Estuquiña point to contact with coastal populations (Supplementary Note 1). On a larger scale, the Muisca of highland Colombia were known to be part of far-reaching trade networks that went beyond the Andean region, facilitating contact between coastal and inland populations[35] (Supplementary Note 1). Thus, if human-to-human transmission did occur, such trade routes would have provided an opportunity for *M. pinnipedii* to be brought inland via human movements. Since *M. pinnipedii* is a member of an animal-adapted clade that diverged from a predominantly human pathogen, a scenario of human-to-human transmission of these ancient *M. pinnipedii* strains could potentially constitute an example of an animal-associated MTBC strain-type re-adapting to the human host, a phenomenon already observed in nosocomial outbreaks of *M. bovis* in

Spain[59,60]. Today, *M. pinnipedii* is known to cause human infection occasionally, but only under circumstances of prolonged exposure to infected animals[19,20].

The ability of *M. pinnipedii* to infect a range of host species, a trait increasingly observed for many MTBC strains[8,19,61,62], coupled with the high diversity of terrestrial mammal species in South America, invites a conceptual exploration of different transmission pathways that may have facilitated its geographic spread. Modern transmission of *M. pinnipedii* from seals to cattle grazing on the New Zealand coastline has been observed[63], as well as transmission among pinnipeds and other captive zoo animals[62]. Additionally, pathogenicity experiments have shown guinea pigs—one of the few domesticated animals in South America—to be susceptible to *M. pinnipedii*[20], as well as to human-adapted MTBC strains[64]. Studies have demonstrated the complexity of *M. bovis* transmission pathways involving a variable range of host species that depend on a region's local ecology[8,13]. It is feasible that animal-to-human transmission could have occurred independent of, or concurrent to, human-to-human transmission, allowing it to spread across South America via a terrestrial route. Dispersal across the geographical expanse considered here would require the involvement of at least one, if not multiple, terrestrial host species able to maintain the infection. Frequent contacts would be required between humans and any non-human host to facilitate transmission, thus making domesticated animals or widespread rodent species potential candidates. In such a scenario, we must also consider that infected humans may have been able to transmit *M. pinnipedii* back to the original and/or other animal hosts. Future study of archaeological faunal material may help clarify the contributions of animal host(s) in past transmission networks; however, the current data are insufficient to elucidate the transmission chain fully and the potential involvement of past animal hosts, or lack thereof.

The western coast of South America has been proposed as the geographic region where human TB infections first developed in the Americas[16]. This is supported by archaeological data that demonstrate Peru and northern Chile to have the highest density and earliest cases of pre-contact human TB-like skeletal lesions[16,17]. Subsequent adaptation to, or maintenance by, humans and/or other terrestrial animal hosts and their interactions via trade routes across the Americas could account for the later emergence of TB-associated skeletal lesions in North American populations that appear, mostly at inland sites, as early as 900 CE[16]. To test this hypothesis, it will be necessary to analyze human skeletal remains from pre-contact North American sites for the presence of strains belonging to the *M. pinnipedii* clade.

To date, three studies have incorporated ancient genome data into Bayesian molecular dating analyses of the MTBC phylogeny, arriving at an emergence date of ~6000 YBP[12,65,66]. If correct, this emergence date precludes the possibility of humans having brought MTBC to the Americas during the initial waves of settlement approximately 15,000 YBP[67]. Transmission via pinnipeds remains the most parsimonious explanation for a route of entry of the pathogen to the Americas from Africa and Eurasia, where MTBC is assumed to have originated[1]. We note that psittacine birds (parrots) kept as pets can on occasion become infected by *M. tuberculosis* through close contact with infected humans[68,69]; however, we consider an avian entry of MTBC to the Americas unlikely, albeit not impossible.

The current phylogeny reveals that our three new ancient human-derived *M. pinnipedii* strains occupy the most basal positions of this clade (Fig. 2). Taken at face value, these data could be interpreted as a human transfer of the pathogen to pinnipeds rather than the reverse. Although the transfer of human-associated MTBC strains to other species is a known phenomenon[61,70], we regard this phylogenetic structure to reflect

sampling biases since all ancient MTBC genomes currently derive from humans and all modern genomes derive from pinnipeds.

Our new Peruvian and Colombian *M. pinnipedii* genomes contribute two new branches to the *M. pinnipedii* clade, both of which are basal to the divergence between the previously published ancient Peruvian and modern *M. pinnipedii* genomes (Fig. 2). The two Colombian genomes form the most basal branch, followed by Peruvian genome 82U. Overall, our genomes extend the known diversity of ancient *M. pinnipedii* strains that circulated in pre-contact South America. Additionally, all six ancient genomes partially overlap within a ~600-year time interval between 1028 and 1640 CE[12] (Results; Supplementary Note 1; Supplementary Data 11). The near contemporaneous presence of paraphyletic strains within the *M. pinnipedii* clade at geographically distant locations could indicate multiple introductions of *M. pinnipedii* to South American humans. Near contemporaneous strain diversity is also observed on a narrower geographic scale in the Osmore River valley, where the basal positioning of the 82U genome to all other ancient Peruvian genomes illustrates the presence of multiple strains (Fig. 2). This strain diversity might reflect multiple individuals having become infected by different *M. pinnipedii* strains circulating in ancient pinniped populations through the repeated exploitation of infected seal tissues. It is important to note that we do not currently know if the three previously published Peruvian genomes (54U, 58U, and 64U) evolved from a common ancestor that was introduced to humans on a single occasion—becoming human-adapted and spreading via human-to-human transmission—or whether this reflects a subset of the diversity evolving in pinniped populations that was transferred to humans on independent occasions. If the strain diversity in the Osmore River valley originated from the same pinniped-to-human introduction event of a single strain, one would expect the 82U genome to occupy a monophyletic clade with the other Peruvian strains, which it does not (Fig. 2). Regarding the phylogeographic distribution observed for the Colombian strains, we believe that one or multiple separate introductions of *M. pinnipedii* from pinniped populations to human and/or terrestrial animal populations is currently the most parsimonious explanation for their spread to these inland locations. Additional genomic data for MTBC strains from the pre-contact Americas will help develop these hypotheses further.

Our metagenomic evaluation shows that non-MTBC mycobacterial DNA was enriched alongside the target ancient MTBC DNA during hybridization capture, and these reads seem to have persisted in our dataset despite multiple analytical approaches to remove them (Results; Methods; Supplementary Figure 1). The enrichment of non-MTBC mycobacterial DNA during capture points to the well-known genetic similarity between environmental mycobacteria and MTBC strains[71]. Importantly, our analyses show that environmentally derived mycobacterial DNA can be a confounding factor in the capture and analysis of MTBC DNA. The genomes in this study were retrieved from skeletal remains directly exposed to soil and moisture, likely accounting for the high percentage of contamination by environmental mycobacterial DNA (Supplementary Note 1). Conversely, the Peruvian *M. pinnipedii* genomes generated by Bos et al.[12] were retrieved from individuals interred in arid stone-lined tombs[72], reducing exposure to soil and moisture, likely accounting for the low ratios of non-MTBC mycobacterial DNA observed in these samples (Results). We urge researchers seeking to analyze MTBC DNA from archaeological samples to assess their data for the presence of non-MTBC mycobacterial DNA to avoid inadvertent misinterpretations that result from analytical artifacts stemming from environmental contamination as opposed to true variation in the MTBC.

In summary, this study extends our knowledge of the phylogeographic and genetic diversity amongst ancient *M. pinnipedii*

strains previously known to be circulating in pre-contact South American populations[12]. By doubling the number of ancient MTBC genomes retrieved from the Americas, we highlight *M. pinnipedii*'s capacity for human infection in unexpected inland locations where contact with marine mammals, or their infected tissues, would have been limited or non-existent. Future research into the genomics of ancient and modern animal MTBC infections will contribute to a better understanding of MTBC's past history and transmission networks in both archaeological and modern contexts.

## Methods

**Sampling and DNA extraction.** DNA was extracted from bone powder sampled from the ribs and vertebrae of nine individuals excavated from six different pre-/peri-contact archaeological sites across Colombia (Supplementary Data 1). All individuals displayed lesions compatible with long-term infection by a member of the MTBC. One bone from each individual was sampled, except for individual 281, where three bones were sampled: one vertebra (281a) and two ribs (281b, 281c).

All samples (except 281a, 281c, and 82) were processed in the cleanroom facilities at Arizona State University (ASU), U.S.A. Debris and dirt were removed from the surface of the bones using a sterilized Dremel tool, and the bones were subsampled. The bone sub-samples were wiped with 10% bleach solution followed by distilled water, and UV-irradiated for 1 min on each side, and subsequently powdered using the 8000 M Mixer/Mill (SPEX). DNA extraction was carried out using a protocol tailored for ancient DNA[73], using between 50 and 100 mg of bone powder for each sample. Extracts were eluted in 100 μl EBT buffer (Qiagen) preheated to 65 °C. An extraction negative control was introduced in each batch of extractions to check for possible contamination introduced during the extraction process. All DNA extracts and extraction negative controls were quantified using the Qubit dsDNA High Sensitivity assay (Life Technologies).

Bone samples 281a and 281c were subsampled in the cleanroom facilities at the University of Tübingen, Germany. A dental drill was used to drill bone powder. ~50 mg of bone powder was extracted[73] per sample, and during the lysis step, samples were rotated for at least 16 h. One negative control and one positive control (bone powder from an ancient cave bear) were included in the extraction batch to control for contamination and extraction efficiency. DNA extracts were eluted in 100 μl of TET (10 mM Tris-Cl, pH 8.0; 1 mM EDTA, pH 8.0; 0.05% Tween-20). Sample 82 was processed in the Tübingen facilities as described in Bos, et al.[12].

**Screening for MTBC DNA.** DNA extracts were screened for the presence of MTBC DNA using quantitative PCR (qPCR) assays and an in-solution hybridization capture.

**qPCR assays.** Undiluted extracts and extraction negative controls were tested for MTBC DNA using three TaqMan (Applied Biosystems) qPCR assays. A 1:10 dilution of each extract was also used to test for the presence of inhibitory substances in the ancient DNA extracts. The first qPCR assay (rpoB2 assay) targets a region of the *rpoB* gene, which is a single-copy gene found in all bacteria and codes for RNA polymerase subunit B. This assay uses a TaqMan probe that binds to an MTBC-specific sequence in the gene[24]; however, due to lack of sequence data for numerous mycobacterial species, this assay might test positive for closely related mycobacterial species as well. The other two assays target regions of the multi-copy insertion elements IS6110 and IS1081 that are found in the MTBC[74–77]. Primer and probe sequences for each qPCR assay are given in Supplementary Data 15. Genomic DNA from *M. tuberculosis* H37Rv (ATCC Catalog Number 25618D-2) was used to create DNA standards for the qPCR assays. Ten-fold serial dilutions ranging from one to 1,000,000 copy numbers of the genome per μl were used to plot a standard curve for quantification purposes. Non-template controls (PCR-grade water, Sigma) were also included on each qPCR plate. DNA extracts, extraction negative controls, and non-template control were run in triplicate whereas DNA standards were run in duplicate. qPCR reactions were run in a 20 μL total volume: 10 μl of TaqMan 2X Universal MasterMix (Applied Biosystems), 0.2 μl of 10 mg/mL RSA (Sigma), and 2 μl of the sample (DNA, standard, or non-template control). Primers and probe were added at concentrations as optimized in Housman et al.[78] (Supplementary Data 15). The qPCR assays were carried out on a 7900HT thermocycler (Applied Biosystems) with the following conditions: 50 °C for 2 min, 95 °C for 10 min, and 50 cycles of amplification at 95 °C for 15 s and 60 °C for 1 min. The results were visualized using SDS v2.3 (Applied Biosystems). Both amplification and multicomponent plots were used to classify the replicates of the extracts as positive or negative. An extract was considered to be positive for a qPCR assay if two or more replicates out of three were positive. qPCR assay results for extracts from samples 281a and 82 have been previously published elsewhere[24].

**In-solution hybridization gene capture.** DNA extracts considered to be positive for one or more qPCR assays were converted into double-indexed libraries[79,80] using 10–20 μl of extract, following the procedure given in Bos et al.[12]. Libraries

were indexed using AmpliTaq Gold (Life Technologies) for 20 cycles and quantified using the DNA1000 assay on the Bioanalyzer 2100 (Agilent Technologies) and the KAPA Library Quantification kit (KAPA Biosystems) using the manufacturer's protocol. A library negative control was included in each batch of samples that underwent library preparation. All libraries, including all extraction and library negative controls, were target-enriched using an in-solution capture protocol at ASU. The libraries were target-enriched for five genes[12,81]: rpoB, gyrA, gyrB, and katG genes commonly found in all mycobacterial species, and the mtp40 segment believed to be unique to certain MTBC strain types, although it is not present in all[82]. Enriched libraries were amplified to a concentration of $10^{13}$ copies per reaction using AccuPrime Pfx DNA polymerase (Life Technologies) and quantified using the Bioanalyzer 2100 (Agilent Technologies) and the KAPA Library Quantification kit (Kapa Biosystems). The libraries were pooled at equimolar concentrations and sequenced on an Illumina MiSeq using V2 chemistry (2×150 bp) at the DNASU Sequencing Center, ASU.

The sequenced reads were trimmed and merged using SeqPrep v1.2 (https://github.com/jstjohn/SeqPrep) using default parameters, except the minimum overlap for merging was modified to 11. Merged reads were mapped to the hypothetical MTBC ancestral reference genome[83] using the Burrows-Wheeler Aligner (bwa v0.7.5)[84]. Bwa-aln was run with seeding turned-off (-l 1000) and a stringent maximum edit distance (-n 0.1) to avoid reads from closely related soil-derived mycobacteria from mapping. Contaminating DNA from soil-derived bacteria is known to be present in archaeological tissues interred in the soil[25]. SAMtools v0.1.19[85] was used to filter the mapped reads at a minimum Phred threshold of Q30, remove duplicate reads and reads that map equally well to more than one position in the genome. The resulting BAM files were visually analyzed using Geneious R7 (Biomatters) to determine the percentage of the targeted genes covered at least one-fold.

**Sample 82.** A sample from individual 82 was previously screened for the presence of MTBC DNA via gene capture and qPCR[12,24], but due to low coverage after gene capture it was not deemed positive enough to warrant whole-genome capture[12]. However, here we chose to include this weak-positive sample, since the probe set used in this study is more specific in its design (described below) than what was previously used by Bos, et al.[12]. Here, we used 60 µl of extract from sample 82, previously generated at the University of Tübingen (refer to Methods in[12]), to make a uracil-DNA-glycosylase (UDG)-treated Illumina library[79,80] that was included in our whole-genome in-solution capture.

**Probe design.** Single-stranded probes for in-solution capture were designed using a computationally extrapolated ancestral genome of the MTBC[83], which was generated by reverting phylogenetically informative positions in the H37Rv reference genome (NC_000962.1) to their ancestral state, hereon referred to as MTBC_anc reference. The probes were 52 bp in length with 5 bp tiling, yielding a set of unique 852,164 probes after the removal of duplicate and low complexity probes. The probe set was raised to 980,000 by random filling of probes. A linker sequence (5′-CACTGCGG-3′) was attached to each probe sequence, resulting in 60-base probes, which were printed on a custom-designed 1 million-feature array (Agilent Technologies). The printed probes were cleaved off the array, biotinylated, and prepared for capture according to Fu, et al.[86].

**MTBC whole-genome in-solution capture and data evaluation.** UDG-treated libraries (U) were generated for the five sample libraries deemed positive for ancient MTBC DNA: 82U, 281aU, 281bU, 281cU, and 386U. Whole-genome in-solution capture[86] was performed for UDG treated and non-UDG (nU) treated libraries, and each library was captured separately in a single well on a 96-well plate. A previously published sample from which an ancient M. pinnipedii genome had been captured (58U)[12] was included in our capture as a positive control. Prior to capture, all sample libraries and negative controls were amplified using Herculase II Fusion DNA Polymerase (Agilent Technologies) to a minimum concentration of 200 ng/µl. The protocol for in-solution capture was carried out according to Fu, et al.[86] with the exception that Cot-1 DNA was excluded from the capture master-mix; therefore 7.5 µl of DNA template was added instead of 5 µl, as listed in the original protocol. The samples were subjected to two rounds of capture. Amplified extraction and library negative controls were pooled and captured together in one well in a separate capture experiment (as above), for a single round.

Initial paired-end sequencing (2x75bp) of the captured samples was carried out on part of an Illumina HiSeq 4000 lane, while extraction and library negative controls were single-end sequenced as part of a HiSeq 4000 lane (1x75bp) (Supplementary Data 2). De-indexing of the sequenced capture data was carried out using bcl2fastq (Illumina; http://support.illumina.com/downloads/bcl2fastq-conversion-software-v217.html). The EAGER pipeline[87] (v.1.92.55) was used to pre-process, map, and estimate damage patterns. Specifically, AdapterRemoval v. 2.2.0[88] was used to clip adapters and merge paired-end reads with a minimum overlap of 11 bp, and only merged reads were kept for downstream analyses. Captured sequence data were mapped to the MTBC_anc reference using bwa-aln (v. 0.7.12)[84], where lenient parameters were used for non-UDG treated libraries (-l 16, -n 0.01) to allow for the mapping of reads with deamination derived mismatches. Stringent parameters were used for UDG treated libraries (-l 32, -n

0.1) where deamination derived lesions had been removed. Negative controls were mapped to the MTBC_anc using lenient parameters, regardless of library treatment. The BAM files were then filtered with SAMTools view using a minimum mapping quality filter (MAPQ; -q) of 37. Duplicate removal was executed using DeDup v. 0.12.2[87] and the assessment of DNA damage patterns with mapDamage 2.0[89].

**Metagenomic analysis of capture and shotgun data.** Shotgun data for both UDG treated and non-UDG treated libraries (excluding 281a and 281c) were generated to evaluate the capture efficiency and the metagenomic profiles of our samples. Libraries were paired-end (2x75bp) shotgun sequenced on part of an Illumina HiSeq 4000 lane or single-end (1x75bp) sequenced as part of an Illumina Next-Seq 500 run (Supplementary Data 2). Pre-processing and mapping of the data were carried out as described above, using stringent mapping parameters for the UDG treated shotgun data. AdapterRemoval v. 2.2.0[88] was used to remove adapters for both paired- and single-end data, and it was also used to merge paired-end reads. Capture efficiencies were calculated based on the number of quality-filtered reads before duplicate removal.

The pre-processed shotgun and capture data were analyzed using the Megan ALignment Tool (MALT) (version 0.3.8)[25]. The shotgun (non-UDG) data previously generated for the three published Peruvian MTBC genomes[12] were also processed with MALT. These data were analyzed using a database comprised of all complete genomes in the NCBI Nucleotide (nt) database downloaded from ftp://ftp-trace.ncbi.nih.gov/blast/db/FASTA/ (7 Dec. 2016) created using malt build (v. 0.3.8). The purpose was to assess the amount of non-MTBC DNA in our shotgun and capture data, particularly with regard to the amount of non-MTBC mycobacterial DNA. Two MALT runs were performed, the first using a minimum percent identity parameter (--minPercentIdentity) of 85, which is a more sensitive alignment criterion. The second run used --minPercentIdentity 95, allowing fewer mismatches in the reads aligned to the database. BlastN mode and SemiGlobal alignment were applied. All other parameters were set to default, except a minimum support parameter (--minSupport) of 1 and a top percent value (--topPercent) of 1 was used. MEGAN6 v.6.12.3[90] was used to view the MALT results. Taxon tables of the MALT results for the shotgun and capture data are shown in Supplementary Data 3, 4, 5, 6, 7 and 8. The captured negative controls were also analyzed with MALT as described above using --minPercentIdentity 95 (Supplementary Data 8). Prior to the MALT analysis described above, identical reads were removed from all data (samples and negative controls) using the rmdup function in seqkit v.0.11.0[91]. This was particularly relevant for the negative controls due to the high duplication rates observed after mapping (Supplementary Data 2).

The MALT results for the UDG treated shotgun and capture data were used to generate pie-charts illustrating the ratio of reads in our samples assigned to the MTBC versus non-MTBC mycobacterial taxa present in our samples before and after capture (Supplementary Figure 1; Supplementary Data 3). The MALT results for the non-UDG treated shotgun data for our samples and the previously published Peruvian dataset[12] were used to compare these ratios (Supplementary Data 5, 6 and 8).

**Deep sequencing, reference-based mapping, and phylogenetic analysis.** Based on the results from the initial sequencing of the capture products, we sequenced the captured UDG treated libraries deeper for samples 82U, 281aU, 281bU and 386U to generate high-coverage genome data. Two libraries, 281aU and 281bU, originate from a vertebra and a rib respectively, from the same individual. An additional captured library generated from another rib (281cU) from this same individual was not sequenced further, because it was determined to contain higher levels of contaminating environmental DNA and lower amounts of endogenous ancient MTBC DNA (Supplementary Figure 1, Supplementary Data 3 and 4). Deeper sequencing was carried out as part of several single-end Illumina HiSeq 4000 (1x75bp) sequencing runs.

The additional single-end sequencing data were adapter-clipped and quality filtered, excluding reads shorter than 30 bp. These data were combined with the merged and quality filtered paired-end data that had already been generated. The combined single- and paired-end data files were treated as single-end data for downstream analyses. MarkDuplicates (Picard tools, v.1.140), which considers only the 5-prime end of reads, was used for duplicate removal, because the true 3-prime end is often not observed in single-end sequences. The combined data for each library were in all other respects subjected to the same mapping and variant calling pipeline as described above, where stringent parameters were used when mapping with bwa-aln to the MTBC_anc reference. Variant calling was also executed with the EAGER pipeline during the processing of the deeper sequenced data using the GATK UnifiedGenotyper (v. 3.5)[92] where the 'EMIT_ALL_SITES' function was activated, providing a call for all variant or non-variant bases in the vcf file.

The dataset of ancient and modern MTBC genomes compiled by Bos, et al.[12] was used in this study for phylogenetic comparison, with the exception of the ancient Hungarian genome that is a composite of two strains[93]. The same vcf files previously generated for the 258 modern genomes were used in this study[12]. The capture data for the three ancient Peruvian individuals were reprocessed using our pipeline in the same manner as our ancient data. Two recently published modern animal-associated MTBC genomes, M. microti strain 12 and one M. mungi were also included in our dataset[37,38]. The assembled genome for M. microti strain 12

(accession no. CP010333.1; https://www.ncbi.nlm.nih.gov/nuccore/CP010333.1) was downloaded and converted into artificial read data (100 bp reads with 1 bp tiling) and subsequently mapped to the MTBC_anc reference as single-end data. The paired-end *M. mungi* BM22813 (accession no. SRS1434640) data were concatenated and mapped as single-end data.

A SNP alignment consisting of only homozygous positions for our four ancient genomes (82U, 281aU, 281bU, 386U), 261 ancient and modern genomes from Bos, et al.[12], *M. microti* strain 12 and *M. mungi* BM22813 was generated using MultiVCFanalyzer (v. 0.87)[12] (https://github.com/alexherbig/MultiVCFAnalyzer). Homozygous positions were typed at positions where 90%, or more, of the reads covering a position agreed and that were supported by a minimum of 3 reads. SNPs located in highly variable regions, or regions susceptible to cross-mapping from other taxa, were excluded. We excluded regions as outlined by Bos, et al.[12], specifically: repeat regions, phage-associated genes, *PE/PPE/PGRS* genes, insertion and mobile elements, *muturase* and *resolvase* genes, *REP* family genes, as well as *tRNAs* and *rRNAs*. The resulting SNP alignment file was used to construct an initial Neighbour-joining tree with 1000 replicates using MEGA6[94]. All missing and ambiguous data were excluded.

The genomes generated from 281aU and 281bU appeared to be identical in the initial phylogeny (Supplementary Figure 2). This was confirmed by visual inspection of the 492 variant positions shared between the 281aU and 281bU genomes where a base call could be made for both genomes (Supplementary Data 13). The sequence data from the two libraries were combined for all remaining analyses and are collectively referred to as 281U.

A new SNP alignment was generated, now including the combined 281U library data yielding a higher average coverage for this sample. SNP calling with MultiVCFAnalyzer was repeated and homozygous variant calls were made as described above, with the exception that a SNP required a minimum of 5 reads covering the position. This SNP alignment file was used to build a Maximum Likelihood phylogeny in RAxML v.8[95] using the GTR gamma model with 1000 bootstraps (Supplementary Figure 3). SNPs occurring in variable and cross-mapping susceptible regions were excluded, as described above, and all positions with missing data, for one or more genomes, were excluded from the dataset.

The allelic frequencies of multiallelic positions were investigated due to the longer branch lengths observed for our Colombian and Peruvian genomes compared to the previously published contemporaneous Peruvian genomes[12]. Multiallelic variants were called at all positions with a minimum of 5-fold coverage and a two-state allele frequency between 10–90%. The distribution of SNP allele frequencies for all six ancient samples is shown in Supplementary Figure 4.

To investigate whether the reads covering homozygous SNPs in the terminal branches of the *M. pinnipedii* genomes might result from the presence of non-MTBC DNA in our dataset, we retrieved all reads covering the homozygous SNPs unique to each of genomes 82U, 281U and 386U, as well as those shared only between 281U and 386U. Unique or shared positions were ascertained based on a comparison between all ancient and modern *M. pinnipedii* genomes and the two modern *M. microti* genomes using MuliVCFAnalyzer as described above for homozygous SNP calling. Through this process, we identified 138, 101 and 110 unique SNPs for 82U, 281U and 386U respectively, and 45 shared SNPs between 281U and 386U for further investigation. All reads covering these SNPs were extracted from the BAM files after duplicate removal using SAMTools *view*. *bam2fq* was used to convert them to FASTQ format, followed by seqtk (v 1.3; https://github.com/lh3/seqtk) (seq -A) to convert them to FASTA format. The read sequences were used as queries in a BLASTN[39] search against the complete NCBI-nt database (April, 2021) with the following parameters: e-value = 1e-6, max_target_seqs = 5. Where one or more of the top five matches was not MTBC, the read was removed from the BAM file using FilterSamReads (Picard tools, v.1.140). SNP calling was repeated, and a Maximum Likelihood tree re-built, as described above, with 1000 bootstraps (Supplementary Figure 5), as well as a Maximum Parsimony tree in MEGA6[94] with 500 bootstraps (Fig. 2). In total, 38 homozygous variant calls were removed. 26, 3, and 8 unique positions were removed from genomes 82U, 281U and 386U, respectively, as well as one position shared by 281U and 386U. Only one of these variant calls led to a change in the full phylogeny (Fig. 2) where complete deletion (exclusion of all missing data) was applied. This single position was removed from the terminal branch of genome 82U.

**Bayesian-based molecular dating using BEAST.** A SNP alignment of all modern and ancient *M. pinnipedii* genomes and *M. microti* genomes was generated with MultiVCFanalyzer, as described above, for homozygous SNP calling at 5-fold coverage. This SNP alignment was used for Bayesian-based molecular dating analysis. Sites with missing or ambiguous data were removed, resulting in a total of 775 sites. To assess whether there was a sufficient temporal signal in the data, a regression of root-to-tip genetic distance against dates of the MTBC strains was conducted using TempEst v1.5.1[96]. The R² value calculated in TempEst was 0.8077, signifying a positive correlation between genetic divergence and time for the *M. pinnipedii-M. microti* strains. Therefore, these data were found to be suitable for molecular clock analysis.

The dating analysis was conducted using BEAST v1.8.4[40]. The calibrated radiocarbon dates of the ancient strains were used to calculate mean years before

present (YBP, with the present being considered as 2020) and were used as priors. For the modern *M. pinnipedii* and *M. microti* strains, a fixed date of 10 YBP was used. The prior for the substitution rate was specified as a normal distribution, with a mean of $4.63 \times 10^{-8}$ ($3.03 \times 10^{-8}$ to $6.21 \times 10^{-8}$ 95% HPD) substitutions per site per year, as estimated by Bos et al.[12]. The two *M. microti* genomes were specified as outgroups. Using jModelTest2[97], the transversional substitution model (TVM) was determined to be the best model of nucleotide substitution. To account for ascertainment bias that might result from using only variable sites in the alignment, the estimated numbers of constant As, Cs, Gs, and Ts were included in the analysis.

We tested a combination of clock and tree models in BEAST, including strict clock and uncorrelated relaxed clock with lognormal distribution (UCLD) models, and coalescent constant population size, coalescent exponential growth, and Bayesian Skyline demographic models. For model testing, one Markov Chain Monte Carlo (MCMC) run was carried out with 100,000,000 iterations, sampling every 10,000 steps. The first 10,000,000 iterations were discarded as burn-in. Maximum Likelihood Estimation (MLE) was conducted using path sampling/stepping-stone sampling in BEAST, over 10,000,000 iterations and 100 path steps, sampling every 1,000 iterations. Tracer v.1.7.1[98] was used to visualize the results of the MCMC runs. Based on the log-likelihood values, the model assuming a UCLD clock and a coalescent constant population size was found to be the best model.

Using the UCLD clock and coalescent constant population size demographic model, we performed a further two independent MCMC runs, each at 1,000,000,000 iterations, sampling every 10,000 steps. The results of all three runs using this model were combined using LogCombiner v1.8.4[99]. The first 10% iterations of each run were discarded as burn-in, resulting in a total of 1,890,000,000 iterations being analyzed. Tracer was used to verify within- and between-chain convergence and that the effective sample size (ESS) values were sufficient. TreeAnnotator v1.8.4[99] was used to summarize the information from the sample of trees produced onto a single target tree calculated by BEAST. Figtree v.1.4.3 (http://tree.bio.ed.ac.uk/software/figtree/) was used to visualize the Maximum Clade Credibility (MCC) tree with mean heights (Supplementary Figure 6).

**Deamination rates analysis.** The deamination rates observed for the non-UDG treated MTBC capture sequences were low, especially for 82nU (Supplementary Data 2). To exclude cross-mapping reads from closely related soil-dwelling mycobacteria from artificially lowering the true deamination rates, we clipped 4 bp off the 3-prime and the 5-prime ends of all reads in the non-UDG data, to remove the bases most likely affected by deamination for samples 82nU (combined 281anU and 281bnU) and 386nU. This was done using seqtk (v 1.3) (trimfq -b -e). The clipped reads were mapped to the MTBC_anc reference with bwa-aln using stringent parameters (-l 32, -n 0.1, -q 37). The IDs for the stringently mapped reads were used to extract the corresponding non-clipped reads from the non-UDG treated sequence data. Mapping with bwa-aln to the MTBC_anc reference with lenient parameters and mapDamage 2.0[89] were subsequently repeated. Results are shown in Supplementary Data 9.

The non-UDG MTBC whole-genome capture data were also mapped to the human genome reference (hg19; NCBI GenBank accession: GCA_000001405.27; https://www.ncbi.nlm.nih.gov/assembly/GCF_000001405.38/) using bwa-aln with lenient parameters (-l 16, -n 0.01), with a MAPQ 37 filtering. Deamination patterns were determined using mapDamage 2.0[89] for the human reads that were co-enriched during the MTBC capture (Supplementary Data 10).

**SNP analysis of protein-coding genes.** SNP analysis was carried out for a subset of genomes selected from the full dataset that was used in phylogenetic analyses. This subset included all ancient and modern *M. pinnipedii* genomes, two *M. microti* genomes and three human adapted *M. tuberculosis* Lineage 6 (L6) genomes, which functioned as outgroups during SNP filtering (see Supplementary Data 13). A SNP table of homozygous variant positions called at a minimum of 5-fold coverage where 90% or more reads supported a base call was generated using MultiVCFanalyzer. The variant calls were annotated using SnpEff v. 3.1[100] with standard parameters. A custom annotation database for non-protein-coding and protein-coding genes in the H37Rv genome was used[12], because this was the genome for which the design of the hypothetical ancestral MTBC genome was based on[83]. An annotated SNP table is shown in Supplementary Data 13.

**Regions of difference in animal lineages.** Regions of difference reported for animal-associated MTBC strains (RD2seal, RD7, RD8, RD9, RD10 and RDmic) were investigated in our three ancient genomes. The merged reads from the capture data were mapped to the MTBC_anc reference genome with bwa-mem[84] using standard bwa-mem parameters in the EAGER pipeline, including all post-mapping steps as previously described in the Methods. BAM files (duplicate reads removed) were visually inspected in IGV v2.8.0[101] and the number of reads spanning the deletion breakpoints were counted. Results are presented in Supplementary Data 14.

**Reporting summary**. Further information on research design is available in the Nature Research Reporting Summary linked to this article.

## Data availability

All raw sequencing data for the MTBC-genome captured libraries which yielded the three *M. pinnipedii* genomes analyzed in this study (samples 82, 281, and 386) and captured negative control libraries have been deposited in the NCBI Sequence Read Archive under BioProject accession number PRJNA779792. The final contaminant filtered BAM files used for variant calling are available on GitHub (https://github.com/ashildv/South_American_ancient_M.pinnipedii). Public datasets used in this manuscript include previously published MTBC genomes for which all accession numbers are provided in Supplementary Data 12, the human reference genome (hg19) available under NCBI GenBank accession code GCA_000001405.27 (https://www.ncbi.nlm.nih.gov/assembly/GCF_000001405.38/) and the full NCBI Nucleotide collection (nt) database downloaded on 7th Dec. 2016 (https://ftp-trace.ncbi.nih.gov/blast/db/FASTA/).

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

## Acknowledgements

We thank Instituto Colombiano de Antropología e Historia (ICANH) and Instituto Nacional de Cultura del Perú (INC) for granting access to the samples and supporting this research. A permit for the Colombian samples was obtained from Autorización de Intervención Arqueológica (permit number 5304). We are immensely grateful to Fernando Montejo Gaitan who supported the authorization procedures in Colombia and to Sloan Williams, Nikki Clark, Don Rice and Geoffrey Conard who were part of the excavations at Estuquiña. We thank Ana Maria Boada, Jose Vicente Rodríguez Cuenca, Pedro Maria Arguello Garcia, Patricia Ramirez Nieto, Maria Angelica Garcia and Juan David Hernández Restrepo who supported this project at various points along the way. We are grateful to Guido Brandt for assistance with laboratory work, Michelle O'Reilly

for graphical support, Ron Hübler for technical support and Elizabeth Nelson for thoughts and discussions on the manuscript. Funding was provided by the Max Planck Society (J.K., K.I.B.); European Research Council Starting Grants APGREID (J.K.) and CoDisEASe (805268) (K.I.B.); Social Sciences and Humanities Research Council of Canada postdoctoral fellowship grant (756-2011-501) (K.I.B.); National Science Foundation (BCS-1063939) (A.C.S., J.E.B) (BCS-1515163) (A.C.S, J.E.B, M.S.R); and The Wenner Gren Foundation for Anthropological Research (A.C.S).

## Author contributions

A.C.S., J.E.B, J.K., and K.I.B. conceived the investigation. A.H., K.I.B., T.P.H., Å.J.V., A.C.S., and J.K. designed experiments. J.E.B., F.C.A., L.P.L., and J.A. identified samples for analyses and provided archaeological information. Å.J.V., T.P.H., K.I.B., K.M.H., and K.G. performed laboratory work. Å.J.V., T.P.H., A.H., M.S.R. and K.I.B. performed analyses. Å.J.V., K.I.B., and T.P.H. wrote the manuscript with contributions from all co-authors.

## Funding

## Competing interests

The authors declare no competing interests.
