## [Peer Review File · Nature Communications]

Geographically dispersed zoonotic tuberculosis in pre-contact South American human populationsReviewers' Comments:

Reviewer #1:

Remarks to the Author:

Vågene et al. present 3 new MTBC genomes from human remains in coastal Peru and inland Colombia, dating from the pre- or peri-contact period with Europeans. This work extends a previous study by Bos et al. (2014) that focused on ancient coastal Peruvian remains, where MTBC genomes were phylogenetically assigned to *M. pinnipedii* and suggested a zoonotic pinniped transmission.

The authors screened 8 individuals from Colombia using qPCR, and included results from 2 individuals (1 from Columbia, 1 from Peru) screened previously by Harkins et al. (2015). Overall, only 4 individuals (82, 281, 382, and 386) were positive for MTBC, including the 2 already reported in Harkins et al. (2015).

After a more in-depth screening of the 4 qPCR-positive individuals using capture of MTBC genes *rpoB*, *gyrA*, *gyrB*, *katG* and *mtp40*, one individual (382) was discarded. The remaining 3 individuals (2 from Columbia 281 and 386, 1 from Peru 82) were used for whole genome capture and downstream analyses.

The authors adopted a very careful approach to screen samples and monitor DNA contaminants, and they investigated carefully the potential impact of contamination on their results. Moreover, the authors acknowledge the presence of contaminants (p.9: "This demonstrates that our novel libraries contain a substantial amount of non-MTBC mycobacterial DNA background in comparison to the samples published by Bos et al., and that these non-target reads persist after stringent (95% ID) filtering."). However, I still have a few queries regarding contamination with exogenous MTBC and non-MTBC DNA, which should motivate a more detailed discussion of the potential origin of the contaminants:

1- Supplementary Table 1 does not report results for extraction and qPCR blank controls. Why?

2- The authors point to differences in the archaeological context (burial in soil versus stone tombs) to explain the presence of more exogenous contaminants than in Bos et al. (2014), but I think they should discuss the possibility of laboratory contaminants based on the following observations:

2a- Supplementary Table 2: non-UDG extraction and library blank controls have a high endogenous content relative to the samples, even if the sequencing effort was limited and the resulting number of mapped reads is low. In any case, these results could mean that there is a significant number of MTBC and closely related non-MTBC DNA fragments in the blank controls.

2b- some discrepancies in 82nU results in Supplementary Tables 2 and 6 show that in this study sample 82 has less merged reads (i.e. a higher content in reads longer than 300 bp that are potential contaminants) and reads are 5 bp longer on average than in Bos et al. (2014). It looks like contaminants have crept up between the two studies, or the differential contamination can be explained by the use of different laboratories between the two studies. It is actually not clear if all the laboratory work beyond the preparation of samples and DNA extraction were performed at ASU in the present study, as opposed to ASU and Tuebingen in Bos et al. (2014).

3-MALT analysis has not been performed on blank controls, or at least results are not reported in Supplementary Table 4. I think the MALT analysis should be performed on the blank controls (in addition to reads mapping) to compare the proportion of non-MTBC mycobacterial DNA background in blanks and samples. It may indicate that contaminants come from library/sequencing reagents or a sub-optimal design of the capture baits (although the baits were used successfully previously), rather than the soil where the samples were buried.

4- While scrutinising the damage results, I noticed a few discrepancies between the text and results in Supplementary Table 2. For example, 82nU has 0.047 for the first base of the 5' end of the reads in

Supplementary Table 2, but 4.16% in the text; the average for 281anU (0.102) and 281bnU (0.077) is 0.0895 in Supplementary Table 2, but it is 8.58% in ms; 386nU is okay.

5- Did the authors try to use PMDtools (Skoglund et al. 2014) on the nU data to retain reads that contain damage, map them, and compare the endogenous content?

6- p.9: "Our extraction and library negative controls did not contain any MTBC DNA after in-solution capture". Results in Supplementary Table 2 clearly show that this statement is wrong. In fact, endogenous content for all blank controls is relatively high when compared to samples. I strongly suggest editing the sentence p.9.

Other major comments not related to contamination include:

7- p.21: "Two studies to date [...] contextualize our findings." I fail to understand the relevance of this paragraph for the present study since the authors did not perform any molecular dating analysis, nor did they use or estimate a substitution rate. I suggest removing the paragraph entirely, but if the authors want to keep it they may want to discuss why they could not perform a Bayesian analysis given the likely impact of exogenous contaminants on variant calls and resulting phylogenetic branch length.

8- Is the deletion characteristic of *M. pinnipedii* (as opposed to *M. microti*) present in the new ancient MTBC genomes?

9- I find the functional description of SNPs (pp. 14-16) rather lengthy and relatively pointless as it is presented. Indeed, the authors write in the discussion that "The functional implications of the SNPs identified by our study are unknown but could be the result of selective pressures." (p.22) In my opinion, it is far too speculative to jump from unknown implications to selection in the same sentence. Why didn't the authors perform positive selection analyses like for *ctpA* in Bos et al. (2014)? At least such analyses would provide substantial evidence to test the hypothesis of selective pressures.

10- At face value, and assuming that the number of substitutions along the new MTBC lineages is not too inflated due to contamination, there seems to be a correlation between genetic and geographic distance if we consider a zoonotic infection site restricted to the estuary of the Osmore River. I fail to reconcile the topology of the tree with isolation by distance, but is it possible to discuss IBD in a human-to-human transmission scenario?

Minor comments:

11- Discrepancy for archaeological ID of MTBC-positive individual 281: LD-90-1X-11 in Harkins et al. (2015), LD-X-011 here.

12- p.6: "endogenous DNA content ranging from 0.95% to 2.08% (Supplementary Table 2)". These reported values correspond to endogenous content calculated with all reads, but values reported in Table 1 are with quality filtered reads only. I suggest reporting in the text only the endogenous content after quality filtering and refer to Table 1 instead of or in addition to Supplementary Table 2.

13- pp.6-7: "The UDG-treated captured library for 281cU did not meet our threshold of 0.4% MALT-assigned endogenous MTBC reads after capture." Could the authors explain why they chose this threshold?

14- p.27: "A sample from individual 82 was previously screened for the presence of MTBC DNA via gene capture and qPCR, but did not meet the previously set requirements for being included in whole

genome capture." Why is it included in this study then? Please be more explicit about the changes in the requirements leading to the inclusion of 82 in the present study.

Reviewer #2:

Remarks to the Author:

In this study, the investigators followed up on a previous observation that *Mycobacteria pinnipedii* was isolated from ancient human skeletal remains obtained from specimens that predate the Columbian arrival in the "New world." The significance of the original finding lay in the demonstration that a zoonotic mycobacterium may have been responsible for cases of a tuberculosis-like disease prior to the introduction of modern European TB strains to the area. This study expands on that observation by noting the presence of *Mycobacterium pinnipedii* in several more pre-Columbian ancient human skeletons, some of which were found in a region of Columbia that is not coastal, raising the question of whether there are or were other zoonotic sources of transmission of this organism. Notably, *M. pinnipedii* has not been identified in existing TB cases in South America, so even if there were a wide ranging reservoir of zoonotic mycobacteria, its likely that this is no longer involved in transmission to humans. The study identifies several genes that vary between the samples and speculates on the role that these may have played in the evolution of the organism.

I will restrict my comments to the general issues, rather than focus on the methods involved in the analysis of the ancient DNA. The main take home message of the study seems to be that because *M. pinnipedii* was found in remains identified in non-coastal areas, this raises the question of how widespread this organism might have been in animal reservoirs. I have several concerns about this interpretation.

First, it is not clear to me that seal-based *M. pinnipedii* could not have been the source of the infections in the Columbia-based humans. My knowledge of the mobility of the Muisca is non-existent but it certainly does not seem completely improbable that people of the Altiplano visited coastal areas or that seal meat was transported from the coast to these areas. If there are archeological data that suggest this is unlikely, these should be summarized in the paper. But even if *M. pinnipedii* was also present in other animals that might have infected humans or if human to human transmission occurred, I am not convinced that this finding has major implications outside the field of zoonotic mycobacteriology. If *M. pinnipedii* infected guinea pigs in this region, one might expect that it would still be endemic in this population and that some cases would also occur in humans given the widespread distribution of guinea pig in the area.

Secondly, the identification of genes that are variable across these strains is interesting but since there is no observable clinical phenotype associated with these changes, it seems very speculative to try to identify evolutionary pathways.

Finally, the abstract suggests that the paper will address human adaptation of the organism but the scenarios explored are necessarily speculative.

In summary, while it is interesting that *M. pinnipedii* has now been found in skeletal remains that are not restricted to the coastal regions of Peru, I don't think this finding alone will really have a major impact on what is known about the transmission of non-TB mycobacteria in pre-Columbian South American or significantly alters our current conception of the evolution of this species.

Reviewer #3:

Remarks to the Author:

This manuscript introduces additional ancient genomes of the TB causing agent obtained from three pre-columbian human remains from Peru and Colombia. In a phylogeny, these new genomes all cluster with three previously reported genomes from Peru basal to strains isolated from modern pinnipeds.

The manuscript is well written and easy to follow. The analyses, to the degree I can judge, appear sound and carefully conducted using state-of-the-art tools. The challenge in analyzing such data relies in the bioinformatic treatment of the raw sequence data, and I think the way the data was treated here is certainly adequate.

The final outcome of this endeavor is then rather simple: it consists of a maximum parsimony phylogeny of the six available ancient TB strains from South America, along with modern strains isolated from humans and several other species, including a number of pinnipeds. From this the authors then conclude that TB in pre-columbian South America may have spread from pinnipeds to humans, as was previously suggested, but also that human-to-human spread was required, maybe indirectly through domesticated animals, as two of the ancient strains were isolated from remains from Colombia, 600 km from the coast. The conclusion that these Colombian samples were not directly infected by pinnipeds is certainly well supported.

However, the conclusion that humans were infected from pinnipeds in the first place is less clear. The authors conclude this from the basal position of the ancient human lineage to those isolated from pinnipeds. But if the transmission was indeed pinnipeds to humans, then one would expect the strains isolated from humans to fall within the pinniped diversity, not basal to it. Aware of this, the authors argue that the basal position is a result to contamination and / or DNA damage, but it remains unclear why such factors would lead to "mutations" shared among ancient lineages (e.g. 386 and 281).

While I do not claim that pinnipeds are not a potential source, the scarcity of Myobacterium strains analyzed from animal sources does not rule out alternative scenarios. For instance, the closest sister clade to the human / pinnipedia clade consists of strains isolated from rodents. How can the authors rule out that a South American Rodent infected both humans and pinnipedia (potentially via humans)? Or any other unsampled species? Clearly the Myobacterium phylogeny does not reflect the mammal phylogeny, suggesting pervasive horizontal transfer in recent times (as the authors also discuss in the paper).

Since TB is thought to be a human pathogen, I acknowledge that the conclusion of a human to animal and back to human interpretation is not challenged. However, the conclusion that there must have been human-to-human spread in South America during antiquity is based on the very fact that the spread was from pinnipeds initially. Hence, that claim must be corroborated very well by the data, which I feel it is currently not, or at least not given what is presented in the manuscript.

Two small issues regarding the abstract:

- 1) The first sentence, while catchy, has no relation to the manuscript. Please remove it.
- 2) The abstract does not convey that two out of three cases were inland. Adding that information to the abstract would certainly strengthen it.

Please find below responses to the reviewer queries for our manuscript: “*Geographically dispersed zoonotic tuberculosis in pre-contact South American human populations*”.

Major changes:

- Blanks/negative controls were re-captured and re-sequenced
- Our archaeological interpretation about the Estuquiña site in Peru was revised as more information came to light
- All analyses investigating the level of non-MTBC contamination were repeated for the samples and the newly sequenced negative controls, though now with removal of identical/duplicate reads prior to MALT analysis, and results presented in terms of unique reads
- We added the range of the Caribbean monk seal that went extinct in 1952 to Figure 1, panel A

Reviewer #1 (Remarks to the Author):

1– Supplementary Table 1 does not report results for extraction and qPCR blank controls. Why?

Thank you for drawing our attention to this omission. We have added the results for the relevant extraction blanks that were screened along with the samples to Supplementary Table 1. This is also now mentioned in **lines 115-116** of the manuscript.

2– The authors point to differences in the archaeological context (burial in soil versus stone tombs) to explain the presence of more exogenous contaminants than in Bos et al. (2014), but I think they should discuss the possibility of laboratory contaminants based on the following observations:

2a– Supplementary Table 2: non-UDG extraction and library blank controls have a high endogenous content relative to the samples, even if the sequencing effort was limited and the resulting number of mapped reads is low. In any case, these results could mean that there is a significant number of MTBC and closely related non-MTBC DNA fragments in the blank controls.

We thank the reviewer for pointing this out. Upon further inspection of our negative controls it became clear that a library blank belonging to an unrelated project had become highly contaminated by modern TB DNA, and was unfortunately pooled together with the negative controls (extraction and library blanks) for our project during capture and sequencing. Crosstalk between libraries processed together is a known phenomenon for the capture method and sequencing platform we used, and we considered this a likely source for the spurious reads in our negative controls. Since our ancient samples were captured and sequenced separately, we are confident that the phenomenon is restricted to the negative controls.

To rectify this, we re-captured (as before, for MTBC) all negative controls associated with this study and sequenced them to a depth of 374,525- 4,270,686 reads. The new mapping statistics are shown in Supplementary Table 2, and we detect between 38 and 2868 unique reads that map to our MTBC reference when using sensitive mapping parameters. While these numbers are on the order of what we observed in our first set of blanks, the sequencing depth here is higher. Our duplication rate now ranges from 2.246- 94.974, which has increased from 1.00 – 4.54 from the first set. This indicates high redundancy in the new data, consistent with a sequencing depth that approaches saturation.

The shotgun sequenced data from the blanks were de-duplicated (identical reads removed without mapping) and the remaining reads were further analysed with MALT using the full

NCBI Nucleotide (nt) database (as described in **lines 666-668**, Supplementary Table 8). Between 0 and 371 (mean value of 34.7), unique reads were assigned cumulatively (SUM) to the *Mycobacterium tuberculosis* complex and to strains/subspecies at lower taxonomic levels (Supplementary Table 8; row 5425). This is much lower than what we report for our mapping (38-2868 reads, Supplementary Table 2), indicating that a large proportion of these reads derive from non-target sources.

For comparison we also analysed the unique reads from our previous negative controls with MALT. This analysis demonstrates a consistently lower number of unique MTBC MALT-assigned reads in our new dataset, indicating clean working conditions and a lack of cross-contamination this time round. For transparency, a comparison of the results for the two capture batches of negative controls is presented in Response-Table 1 below. As we discuss only the new negative control dataset, this table is not presented in the main manuscript.

Response-Table 1: comparison of MTBC reads from old and re-captured blanks/negative controls

Blank ID	OLD BLANKS						Re-captured BLANKS					
	Total reads (pre-processed with identical reads removed)	Number of unique mapping reads BWA (sensitive parameters)	Number of reads assigned (SUM) to Mycobacterium node in MALT	Number of reads assigned (SUM) to MTBC node in MALT	% MTBC (SUM) reads of Mycobacterium (SUM) reads in MALT	% MTBC (SUM) reads out of total reads (pre-processed) in MALT	Total reads (pre-processed with identical reads removed)	Number of unique mapping reads BWA (sensitive parameters)	Number of reads assigned (SUM) to Mycobacterium node in MALT	number of reads assigned (SUM) to MTBC node in MALT	% MTBC (SUM) reads of Mycobacterium (SUM) reads in MALT	% MTBC (SUM) reads out of total reads (pre-processed) in MALT
EB_020814_AEL9	117785	258	232	95	40.9	0.08066	1318168	294	499	34	6.8	0.00258
EB_020814_AEL29	340868	1366	619	392	63.3	0.11500	3386561	2868	1903	89	4.7	0.00263
EB032016nU_cap	3294	11	1	0	0.0	0.00000	369951	70	34	3	8.8	0.00081
EB032016U_cap	267442	1050	534	443	83.0	0.16564	2013973	1124	382	35	9.2	0.00174
EB032116nU_cap	2987	4	0	0	0.0	0.00000	534517	81	49	1	0.0	0.00019
EB032116U_cap	183713	641	208	145	69.7	0.07893	1589655	1011	241	23	9.5	0.00145
ExtrBlk1_020812_AEL31	55258	285	211	197	93.4	0.35651	165939	129	62	7	11.3	0.00422
ExtrBlk1_030812_AEL33	54831	419	385	351	91.2	0.64015	144295	94	120	5	4.2	0.00347
ExtrBlk2_020814_AEL32	49727	385	340	312	91.8	0.62743	181157	110	119	13	10.9	0.00718
ExtrBlk2_030814_AEL34	57171	398	413	319	77.2	0.55798	168608	110	280	1	0.4	0.00059
ExtrBlk240712_AEL30	33808	183	145	128	88.3	0.37861	106488	90	63	0	0.0	0.00000
LB032416nU_cap	440	17	17	17	100.0	3.86364	166454	57	18	2	11.1	0.00120
LB040116U_cap	29503	267	239	144	60.3	0.48809	312483	342	660	371	56.2	0.11873
LB1_120815_AVL11	11904	90	70	62	88.6	0.52083	63780	38	36	5	13.9	0.00784
LB2_120815_AVL12	13209	122	113	100	88.5	0.75706	82121	39	56	0	0.0	0.00000
LB1150814_AVL35	48653	275	228	215	94.3	0.44190	166089	89	48	1	2.1	0.00060
LB1150814_AVL36	45005	208	177	155	87.6	0.34441	119006	83	65	0	0.0	0.00000

Non-target reads are thought to come either from cross-contamination during laboratory processing, or from other bacteria present in the environment/ reagents. Importantly, no damage pattern could be discerned from any of the non-UDG treated blanks, although reliable damage plots could only be determined for libraries with a minimum of 250 reads.

Regardless, any signals of MTBC DNA that we report in our negative controls are much lower than what we identify in our samples, which further supports the authenticity of our ancient data.

The contaminated and shallow sequenced blanks from our first analysis have been removed from our manuscript and have been replaced by the re-processed set.

2b– some discrepancies in 82nU results in Supplementary Tables 2 and 6 show that in this study sample 82 has less merged reads (i.e. a higher content in reads longer than 300 bp that are potential contaminants) and reads are 5 bp longer on average than in Bos et al. (2014). It looks like contaminants have crept up between the two studies, or the differential contamination can be explained by the use of different laboratories between the two studies. It is actually not clear if all the laboratory work beyond the preparation of samples and DNA extraction were performed at ASU in the present study, as opposed to ASU and Tuebingen in Bos et al. (2014).

In this study, the 82nU (non-UDG) library was the same as that analysed in Bos et al, 2014 ¹, and the 82U (UDG treated) library explored here was made from the same extract. All ancient DNA lab work for sample 82 was carried out in the cleanroom facilities at the University of Tübingen. This has been made clearer in **lines 524 and 590-593** of the main manuscript.

Former Supplementary Table 7 (not 6) showed our mapping statistics for the non-UDG treated shotgun data mapped to the hypothetical ancestor of MTBC that was produced by Bos et al. 2014. This table has now been removed as it was superfluous, but below is a summary of the relevant statistics.

Table 1. Summary mapping stats for sample 82 non-UDG shotgun

Sample 82 non-UDG shotgun reporting source	# reads after adapter clipping and merging (PE) prior mapping	No. merged reads for shotgun	% merged reads	No. reads mapping to TB complex post rmdup	Average read length (number of bases)
Bos et al. 2014	not reported	77640	not reported	629	not reported
This study, former Supplementary Table 7	77437	76884	99.28	72	53.26

The mapping statistics presented for 82nU SG in our former Supplementary Table 7 are the only ones that are based on the same SG data presented in Bos et al 2014. The number of MTBC mapped reads is very different and this is due to the different mapping approaches used between the two studies. The SG data in the Bos et al. 2014 paper was processed with Bowtie2 in local alignment mode, whereas in this study we use bwa-aln (semi-global alignment). In short, these two papers use two different mappers and alignment modes, and are therefore not comparable.

None of the data in Supplementary Table 2 is directly comparable with the 82nU SG data. The UDG treated SG sequenced 82U library has an average read length of 56.3 bases compared to 53.26 bases for the 82nU shotgun. This slight increase is due to the fact that this is based on different numbers of mapping reads 72 (82nU SG) versus 260 (82U SG) due to different sequencing depths. Furthermore, the 82nU capture data shows an average read length of 61.7 bases, which is based on 38,197 mapping reads. It is expected that the average read length will vary slightly between different sequencing depths. Additionally, it is established that capture techniques favor longer fragments. This effect has already been noted for *Yersinia pestis* by Spyrou, et al. ², for example.

With regard to differences in the number of merged reads, this could be due to the sequencing strategy used. 82nU was shotgun sequenced for Bos et al. 2014 using the MiSeq with paired-end 150bp, while the 82U for this study was shotgun sequenced using the HiSeq4000 paired-end 75bp kit. It is, therefore, expected that the reads produced on the MiSeq would have a higher merging rate since it allowed for the sequencing, and thus merging, of longer reads.

Extracts and libraries for sample 82 were processed together with the other samples from Bos, et al. ¹, three from which ‘clean’ genomes were recovered (54U, 58U and 64U). If the contamination was coming from the reagents, then this would also be observed in the 54U, 58U and 64U genomes, which it is not. This, coupled with the MALT analysis and the high duplication rates of the blanks makes it highly unlikely that the background contamination

that we see for the samples in this study came from reagents or other contamination from the lab environment.

3–MALT analysis has not been performed on blank controls, or at least results are not reported in Supplementary Table 4. I think the MALT analysis should be performed on the blank controls (in addition to reads mapping) to compare the proportion of non-MTBC mycobacterial DNA background in blanks and samples. It may indicate that contaminants come from library/sequencing reagents or a sub-optimal design of the capture baits (although the baits were used successfully previously), rather than the soil where the samples were buried.

MALT analysis of the re-captured blanks has been added and is shown in Supplementary Table 8. Due to the high number of duplicate reads in the blanks (see Supplementary Table 2), identical (duplicate) reads were removed before MALT analysis (described in **lines 666-669** of the manuscript). MALT analysis with 95% ID shows 0-371 reads assigned cumulatively (SUM) to the MTBC (this number also includes reads assigned to lower taxonomic levels within MTBC) in the blanks. These are much lower than the number of reads we report as mapping to our chosen reference genome with sensitive parameters. This is a further demonstration that the majority of reads mapping from the blanks are in fact off-target (i.e. not from members of the MTBC).

The probe set used in this study is different from that used in Bos, et al. ¹. In this study probes are based solely on a hypothetical ancestor of MTBC (H37Rv architecture with ancestral SNP alleles, see **lines 595-605** in the main manuscript for further information). In Bos, et al. ¹, the probes were based on genomic diversity found in 21 modern MTBC genomes and the genomes of *M. kansasii* and *M. avium*, which are both genetically divergent from MTBC. Thus, the probes were designed to capture a broader and more diverse set of mycobacterial sequences, and the assay is less suited for capturing MTBC genomes from DNA libraries containing high amounts of mycobacterial diversity from the soil/environment. The probes used in this study more precisely target the MTBC.

4– While scrutinizing the damage results, I noticed a few discrepancies between the text and results in Supplementary Table 2. For example, 82nU has 0.047 for the first base of the 5' end of the reads in Supplementary Table 2, but 4.16% in the text; the average for 281anU (0.102) and 281bnU (0.077) is 0.0895 in Supplementary Table 2, but it is 8.58% in ms; 386nU is okay.

We thank the reviewer for pointing this out. We have amended the text to reflect the correct damage % for 82nU (see **line 203**). For 281nU the 8.58% was derived from the combined non-UDG capture data for 281a and 281b. These mapping statistics have now been added to Supplementary Table 2; they were previously omitted from the table.

5– Did the authors try to use PMDtools (Skoglund et al. 2014) on the nU data to retain reads that contain damage, map them, and compare the endogenous content?

PMDtools was designed to distinguish ancient human DNA from modern human contaminating DNA introduced by people handling the remains during and post excavation. This tool would not be suitable for our study, because we are dealing with contamination of mycobacterial DNA sequences derived from the soil in the environment. The remains would have been exposed to such contamination from the moment the bodies were interred/buried, therefore the contaminating soil mycobacterial DNA will be a mix of ancient DNA (with

damage) and modern DNA. Thus, PMDtools would not be able to distinguish between ancient MTBC DNA and closely related ancient soil-derived mycobacterial DNA.

6– p.9: “Our extraction and library negative controls did not contain any MTBC DNA after in-solution capture”. Results in Supplementary Table 2 clearly show that this statement is wrong. In fact, endogenous content for all blank controls is relatively high when compared to samples. I strongly suggest editing the sentence p.9.

With regard to the analysis explained in our response to comments above we have amended this sentence to read as follows: “Reference-based mapping analyses, using sensitive mapping parameters, revealed as many as 2,868 unique mapping reads in our negative controls, though MALT assigned few of these (between 0 to 371) to the MTBC and lower taxonomic nodes (Supplementary Tables 2, 8). Although the assignment of MTBC reads in non-pathological samples is a known phenomenon ³, these data indicate that the reagents were not the main source of the mycobacterial contaminants observed in our samples.”, in **lines 189-196** of the manuscript.

Other major comments not related to contamination include:

7– p.21: “Two studies to date [...] contextualize our findings.” I fail to understand the relevance of this paragraph for the present study since the authors did not perform any molecular dating analysis, nor did they use or estimate a substitution rate. I suggest removing the paragraph entirely, but if the authors want to keep it they may want to discuss why they could not perform a Bayesian analysis given the likely impact of exogenous contaminants on variant calls and resulting phylogenetic branch length.

We have changed this section considerably to make the purpose of this paragraph clearer, see **lines 421-429** in the manuscript. The purpose of this paragraph was to contextualise the reasoning behind the hypothesis that *M. pinnipedii* strains were introduced to the Americas via pinnipeds.

8– Is the deletion characteristic of M. pinnipedii (as opposed to M. microti) present in the new ancient MTBC genomes?

We thank the reviewer for this comment and have now included analysis of regions of difference in the manuscript. The *M. pinnipedii* (RDseal) region is absent/deleted in the three new ancient genomes presented in this paper. We also investigated these additional regions of difference: RD7, RD8, RD9, RD10 and RDmic. See **lines 312-321 and 781-788** in the main manuscript and Supplementary Table 14.

*9– I find the functional description of SNPs (pp. 14-16) rather lengthy and relatively pointless as it is presented. Indeed, the authors write in the discussion that “The functional implications of the SNPs identified by our study are unknown but could be the result of selective pressures.” (p.22) In my opinion, it is far too speculative to jump from unknown implications to selection in the same sentence. Why didn’t the authors perform positive selection analyses like for *ctpA* in Bos et al. (2014)? At least such analyses would provide substantial evidence to test the hypothesis of selective pressures.*

We have shortened/removed our SNP analyses from both the results and discussion sections at the request of Reviewer 2.

10- At face value, and assuming that the number of substitutions along the new MTBC lineages is not too inflated due to contamination, there seems to be a correlation between genetic and geographic distance if we consider a zoonotic infection site restricted to the estuary of the Osmore River. I fail to reconcile the topology of the tree with isolation by distance, but is it possible to discuss IBD in a human-to-human transmission scenario?

The reviewer is asking us to entertain and discuss a scenario in which the estuary of the Osmore River was the only zoonotic entry point of *M. pinnipedii* to the Americas, and that humans subsequently spread *M. pinnipedii* from Peru to inland Colombia. We do not wish to speculate on such a scenario since there is archaeological evidence supporting the exploitation and consumption of pinniped tissues by Peruvian and Chilean coastal populations, also with evidence in Tierra del Fuego starting in the Pleistocene period as early as 11,000 YBP. By 6000 BP, specialized gatherers and fishers from the southern tip of South America were hunting pinnipeds as their main dietary staple (refer to Supplementary section called “Pinniped Exploitation in Pre-contact South America” in Bos et al. 2014). We therefore believe that it is more likely that *M. pinnipedii* may have been introduced multiple times at multiple geographic regions. We believe that we are only scratching the surface of the genetic diversity of *M. pinnipedii* spanning South and possibly North America since pre-Columbian examples have been identified in Colombia, Venezuela and western Mexico, while the majority of evidence for tuberculosis is found along the coast of modern-day Peru/Chile and in eastern and southwestern North America (refer to Supplementary section called “Paleopathological Evidence of Tuberculosis in the New World” in Bos et al. 2014). The cases in Colombia could have been transported from anywhere along the Peruvian coast, or even possibly the northern coast of Colombia. We have added the range for the extinct Caribbean Monk seal that disappeared in 1952 to Figure 1A, so *M. pinnipedii* could potentially also have been introduced there.

Since the transmission chain is unknown for all ancient ‘*M. pinnipedii*’ genomes, we do not want to speculate further as to how/when/where our ancient strains diverged.

Ultimately, we need more data in order to make inferences about patterns of IBD among ancient *M. pinnipedii* strains in the Americas, assuming human-to-human transmission.

Minor comments:

Discrepancy for archaeological ID of MTBC-positive individual 281: LD-90-1X-11 in Harkins et al. (2015), LD-X-011 here.

The archaeological ID was accidentally shortened and has now been changed to what is written in Harkins et al. 2015 (LD-90-1X-11) in Supplementary Table 1.

12– p.6: “endogenous DNA content ranging from 0.95% to 2.08% (Supplementary Table 2)”. These reported values correspond to endogenous content calculated with all reads, but values reported in Table 1 are with quality filtered reads only. I suggest reporting in the text only the endogenous content after quality filtering and refer to Table 1 instead of or in addition to Supplementary Table 2.

We have amended this in **line 133-140**. In the manuscript, we now refer to the endogenous DNA % based on quality filtered reads only.

13– pp.6-7: “The UDG-treated captured library for 281cU did not meet our threshold of 0.4% MALT-assigned endogenous MTBC reads after capture.” Could the authors explain why they chose this threshold?

0.4% was an arbitrary threshold that was chosen to provide reasoning for the exclusion of 281cU from deeper sequencing. We have changed our approach and now explain it differently in the manuscript (see **lines 152-157**). We have removed the metric threshold and stated that we chose not to continue with 281cU because the two other samples from this individual (281aU & 281bU) had much higher numbers of MALT-assigned endogenous MTBC reads. Therefore, we chose to focus our sequencing efforts on 281aU and 281bU since we would get more on-target reads from these samples.

14– p.27: “A sample from individual 82 was previously screened for the presence of MTBC DNA via gene capture and qPCR, but did not meet the previously set requirements for being included in whole genome capture.” Why is it included in this study then? Please be more explicit about the changes in the requirements leading to the inclusion of 82 in the present study.

Sample 82 was included in this study because we use a more specific probe set, as is now explained in **lines 595-605** of the manuscript. Therefore, we decided to test sample 82 for capture even though it is a weaker-positive sample. The difference in probe sets is also explained in our answer to your point 3 above. The cut-off imposed by Bos et al 2014 was also an arbitrary one.

Reviewer #2 (Remarks to the Author):

*In this study, the investigators followed up on a previous observation that *Mycobacteria pinnipedii* was isolated from ancient human skeletal remains obtained from specimens that predate the Columbian arrival in the “New world.” The significance of the original finding lay in the demonstration that a zoonotic mycobacterium may have been responsible for cases of a tuberculosis-like disease prior to the introduction of modern European TB strains to the area. This study expands on that observation by noting the presence of *Mycobacterium pinnipedii* in several more pre-Columbian ancient human skeletons, some of which were found in a region of Columbia that is not coastal, raising the question of whether there are or were other zoonotic sources of transmission of this organism. Notably, *M. pinnipedii* has not been identified in existing TB cases in South America, so even if there were a wide ranging reservoir of zoonotic mycobacteria, its likely that this is no longer involved in transmission to humans. The study identifies several genes that vary between the samples and speculates on the role that these may have played in the evolution of the organism.*

*I will restrict my comments to the general issues, rather than focus on the methods involved in the analysis of the ancient DNA. The main take home message of the study seems to be that because *M. pinnipedii* was found in remains identified in non-coastal areas, this raises the question of how widespread this organism might have been in animal reservoirs. I have several concerns about this interpretation.*

*First, it is not clear to me that seal-based *M. pinnipedii* could not have been the source of the infections in the Columbia-based humans. My knowledge of the mobility of the Muisca is non-existent but it certainly does not seem completely improbable that people of the Altiplano visited coastal areas or that seal meat was transported from the coast to these areas. If there are archaeological data that suggest this is unlikely, these should be summarized in the paper. But even if *M. pinnipedii* was also present in other animals that might have infected humans or if human to human transmission occurred, I am not convinced that this finding has major implications outside the field of zoonotic mycobacteriology. If *M. pinnipedii* infected guinea pigs in this region, one might expect that it would still be endemic in this population and that some cases would also occur in humans given the widespread distribution of guinea pig in the area.*

We investigated the nitrogen values for the two Colombian individuals to see if we could detect a marine vertebrate component to their diet. For the Colombian individuals included in this study, the d15N values are +9.9 ‰ and +10.2 ‰. This falls within the expected range of human d15N collagen values for ancient maize consumers. If these individuals had a marine vertebrate component to their diet (such as pinnipeds), the expected d15N values would be much higher (between +14 and +15‰) ⁴⁻⁶. Therefore, it is unlikely that these two individuals

(281 and 386) travelled to the coast and consumed infected pinniped tissues. We have made appropriate additions in the manuscript to describe this, see **lines 363-365**.

Secondly, the identification of genes that are variable across these strains is interesting but since there is no observable clinical phenotype associated with these changes, it seems very speculative to try to identify evolutionary pathways.

We have significantly shortened the results section (see **page 15**) about the genetic differences between the strains and we removed the section about specific SNPs in the discussion. It should be noted that we have added a section about the regions of difference that delineate the animal-associated MTBC strains, see **lines 312-321 and 781-788**.

Finally, the abstract suggests that the paper will address human adaptation of the organism but the scenarios explored are necessarily speculative.

We have amended this statement in the abstract, see **lines 50-53**.

*In summary, while it is interesting that *M. pinnipedii* has now been found in skeletal remains that are not restricted to the coastal regions of Peru, I don't think this finding alone will really have a major impact on what is known about the transmission of non-TB mycobacteria in pre-Columbian South American or significantly alters our current conception of the evolution of this species.*

Ancient DNA is a powerful tool in paleopathology, as well as microbial genomics, to explore strain-level designation of past infections. Prior to this work it could be confirmed that MTBC was indeed present in the regions considered here based on skeletal lesions, but identification of the pinniped lineage was restricted to the Osmore River Valley of Peru. Here we both refine our technique of DNA genome reconstruction (demonstrated through the assembly of genome 82, which was considered out of reach in Bos et al 2014) and demonstrate the presence of related MTBC pinniped lineages elsewhere in South America. While our discussions on transmission are highly speculative, and we can offer few insights on the overall evolution of the pathogen species as a whole, we believe both our methodological improvements on the retrieval of genome level data and further characterisation of the enigmatic strains of *M. pinnipedii* in the pre-contact Americas make this a unique and influential contribution to the current literature.

Moreover, MTBC/TB is the most common cause of death due to a single infectious agent today. However, we consider TB, typically, to be a disease of low socioeconomic populations in densely concentrated areas with poor access to healthcare. Our current findings show that in the ancient Americas, a new TB strain (*M. pinnipedii*) was introduced to human populations in Peru on multiple occasions. Regardless of whether the transmission was human-to-human, or animal-to-human, our results demonstrate that *M. pinnipedii* had the ability to spread across long distances 1000 years ago.

Furthermore, our work is highly relevant to research across diverse disciplines that encompass archaeology, history and microbiology.

Reviewer #3 (Remarks to the Author):

This manuscript introduces additional ancient genomes of the TB causing agent obtained from three pre-columbian human remains from Peru and Colombia. In a phylogeny, these new genomes all cluster with three previously reported genomes from Peru basal to strains isolated from modern pinnipeds.

The manuscript is well written and easy to follow. The analyses, to the degree I can judge, appear sound and carefully conducted using state-of-the-art tools. The challenge in analyzing such data relies in the bioinformatic treatment of the raw sequence data, and I think the way the data was treated here is certainly adequate.

The final outcome of this endeavor is then rather simple: it consists of a maximum parsimony phylogeny of the six available ancient TB strains from South America, along with modern strains isolated from humans and several other species, including a number of pinnipeds. From this the authors then conclude that TB in pre-columbian South America may have spread from pinnipeds to humans, as was previously suggested, but also that human-to-human spread was required, maybe indirectly through domesticated animals, as two of the ancient strains were isolated from remains from Colombia, 600 km from the coast. The conclusion that these Colombian samples were not directly infected by pinnipeds is certainly well supported.

However, the conclusion that humans were infected from pinnipeds in the first place is less clear. The authors conclude this from the basal position of the ancient human lineage to those isolated from pinnipeds. But if the transmission was indeed pinnipeds to humans, then one would expect the strains isolated from humans to fall within the pinniped diversity, not basal to it. Aware of this, the authors argue that the basal position is a result to contamination and / or DNA damage, but it remains unclear why such factors would lead to “mutations” shared among ancient lineages (e.g. 386 and 281).

We wish to make clear that we do not dispute the topology of our constructed tree. We believe that the basal split of our genomes from those published in 2014 is correct. We do, however, question the length of the terminal branches of our genomes: 82, 281 and 386, due to the influence of contamination from either environmental mycobacteria that have genetic similarity to MTBC and/or other contaminant DNA sequences that we cannot filter out, even when using a stringent mapping approach (see **lines 272-280**). Further to this, DNA damage does not factor into our reasoning since the data that gave rise to the phylogeny derived from enzymatically repaired libraries (UDG treatment).

We do not expect the past diversity of *M. pinnipedii* to cluster together with modern *M. pinnipedii*. One cannot make inferences about past diversity that may, or may not, exist today. Animal-associated MTBC are poorly studied and the surface of the genomic diversity that exists today, let alone what existed in the past, has barely been scratched. As we state in the manuscript we currently only have ancient *M. pinnipedii* strains derived from humans. Therefore, we have no information on the diversity of strains that were circulating amongst ancient pinnipeds.

We have added more to the discussion about possible transmission scenarios to **lines 367-436** in the manuscript.

While I do not claim that pinnipeds are not a potential source, the scarcity of Myobacterium strains analyzed from animal sources does not rule out alternative scenarios. For instance, the closest sister clade to the human / pinnipedia clade consists of strains isolated from rodents.

How can the authors rule out that a South American Rodent infected both humans and pinnipedia (potentially via humans)? Or any other unsampled species? Clearly the Myobacterium phylogeny does not reflect the mammal phylogeny, suggesting pervasive horizontal transfer in recent times (as the authors also discuss in the paper).

The reason why we believe a transmission from pinnipeds is most parsimonious at this time is that the molecular dating using ancient strains, produced by three different studies, all point to an emergence of ~6000 YBP of the MTBC, meaning that MTBC emerged after the Bering land bridge to the Americas closed in 15,000 YBP. If one assumes these dates to be correct, then there is no way for MTBC to have initially entered the rodent population, or another terrestrial species, unless it was brought to the Americas via a marine or potentially airborne route (see **lines 421-429, 367-436** in the manuscript).

Since TB is thought to be a human pathogen, I acknowledge that the conclusion of a human to animal and back to human interpretation is not challenged. However, the conclusion that there must have been human-to-human spread in South America during antiquity is based on the very fact that the spread was from pinnipeds initially. Hence, that claim must be corroborated very well by the data, which I feel it is currently not, or at least not given what is presented in the manuscript.

The current version of the manuscript now accommodates other interpretations (see **lines 367-436**).

Two small issues regarding the abstract:

1) The first sentence, while catchy, has no relation to the manuscript. Please remove it.

It has been removed.

2) The abstract does not convey that two out of three cases were inland. Adding that information to the abstract would certainly strengthen it.

We have revised our interpretation of Estuquiña (Peru) as a coastal site and now designate it as an inland site throughout the manuscript. The original designation of the site as coastal was due to confusion regarding the archaeological context of the site (see Supplementary section 1).

- 1 Bos, K. I. *et al.* Pre-Columbian mycobacterial genomes reveal seals as a source of New World human tuberculosis. *Nature* **514**, 494-497, doi:10.1038/nature13591 (2014).
- 2 Spyrou, M. A. *et al.* Analysis of 3800-year-old *Yersinia pestis* genomes suggests Bronze Age origin for bubonic plague. *Nat Commun* **9**, 2234, doi:10.1038/s41467-018-04550-9 (2018).
- 3 Warinner, C. *et al.* A Robust Framework for Microbial Archaeology. *Annu Rev Genomics Hum Genet*, doi:10.1146/annurev-genom-091416-035526 (2017).
- 4 Slovak, N. M. & Paytan, A. Fisherfolk and farmers: Carbon and nitrogen isotope evidence from Middle Horizon Ancón, Peru. *International Journal of Osteoarchaeology* **21**, 253-267, doi:10.1002/oa.1128 (2011).
- 5 Schoeninger, M. J., DeNiro, M. J. & Tauber, H. Stable nitrogen isotope ratios of bone collagen reflect marine and terrestrial components of prehistoric human diet. *Science* **220**, 1381, doi:10.1126/science.6344217 (1983).
- 6 DeNiro, M. J. Stable Isotopy and Archaeology. *American Scientist* **75**, 182-191 (1987).

Reviewers' Comments:

Reviewer #1:

Remarks to the Author:

The authors have addressed all my comments very thoroughly. They also generated new data (re-capture and re-sequencing of the blank and negative controls) and performed additional analyses (MALT) that alleviate my concerns over contamination. To be honest I struggle with the concept of processing negative controls separately from the samples because it defeats the purpose of having negative controls in the first place, but it is clear in Supplementary Table 2 that the samples contain mostly authentic ancient MTBC DNA.

I have no further concerns about the manuscript and recommend publication.

Reviewer #3:

Remarks to the Author:

This manuscript introduces three additional ancient genomes of the TB causing agent obtained from three pre-columbian human remains from Peru and Colombia. In a phylogeny, these new genomes all cluster with three previously reported genomes from Peru basal to strains isolated from modern pinnipeds. As these samples were obtained somewhat far from the coast, the first version of this manuscript took this as evidence for ancient human-to-human transmission of TB in South America. As several reviewers pointed out, the presented data does not rule out a number of alternative scenarios.

The authors do not provide any additional data or analysis to substantiate their initial claim. Instead, the revised manuscript now acknowledges that additional samples are required to make any strong statement about ancient TB transmission. Provocatively summarized, the paper therefore simply reports that TB was also present in South America beyond coastal regions. While not an expert on TB, I fail to see the major implication of this finding without further samples or analyses

Reviewer #4:

Remarks to the Author:

The manuscript is well written and easy to follow. The analyses appear to be carefully conducted. The findings of additional ancient *M. pinnipedii* genomes that recovered from human remains in inland sites in Peru and Colombia indicates an additional mode of human-to-human TB transmission and geographic dispersal of TB disease in the pre-contact era in Southern America. The author's responses to the reviewers' comments are appropriate. The reviewer has one additional concern:

Based on current results, significant contamination was suspected on the three samples and this contaminant reads likely to arise from non-MTBC mycobacteria, which likely accounts for the unexpectedly long terminal branch-lengths for the three ancient samples in the constructed phylogenies trees (Fig. 2; Supplementary Fig. 2, 3,4). These caveats also lead to a lack of Bayesian-based molecular dating analysis in this study. However, these non-targeted reads have passed the 90% homozygosity threshold and also lead to shared variants among at least two or three ancient samples, and thus could an attempt be possible to harvest the reads harboring those variants in the long terminal branch of these three samples, and to verify whether these reads belong to non-MTBC mycobacteria or taxa (e.g. Blast)?

This alternative analysis is important and can provide evidence for the real branch length of these three genomes and enable the dating analysis which can add valuable insights into the common ancestor while compared to the previous finding in Bos, K. I. et al. (Nature, 2014). Such unexpected

accumulation of genomic diversity, if it can be verified, can provide a different story on the estimation of the most recent common ancestor for the MTBC.

Meanwhile, such analysis may also contribute to the argument raised by reviewer #3 about the concluded 'pinnipeds-to-humans' transmission route. It is still less clear why the previous ancient samples published in 2014 form a sister branch with the modern *M. pinnipedii* genomes while the current three samples were located to a basal position if these samples resulted from the human(or animal)-to-human transmission after the pinnipeds-to-humans transmission.

RESPONSE TO REVIEWER COMMENTS

Reviewer #1 (Remarks to the Author):

The authors have addressed all my comments very thoroughly. They also generated new data (re-capture and re-sequencing of the blank and negative controls) and performed additional analyses (MALT) that alleviate my concerns over contamination. To be honest I struggle with the concept of processing negative controls separately from the samples because it defeats the purpose of having negative controls in the first place, but it is clear in Supplementary Table 2 that the samples contain mostly authentic ancient MTBC DNA.

I have no further concerns about the manuscript and recommend publication.

REPLY: We thank the reviewer for their comments.

Reviewer #3 (Remarks to the Author):

This manuscript introduces three additional ancient genomes of the TB causing agent obtained from three pre-columbian human remains from Peru and Colombia. In a phylogeny, these new genomes all cluster with three previously reported genomes from Peru basal to strains isolated from modern pinnipeds. As these samples were obtained somewhat far from the coast, the first version of this manuscript took this as evidence for ancient human-to-human transmission of TB in South America. As several reviewers pointed out, the presented data does not rule out a number of alternative scenarios.

The authors do not provide any additional data or analysis to substantiate their initial claim. Instead, the revised manuscript now acknowledges that additional samples are required to make any strong statement about ancient TB transmission. Provocatively summarized, the paper therefore simply reports that TB was also present in South America beyond coastal regions. While not an expert on TB, I fail to see the major implication of this finding without further samples or analyses

REPLY: We thank the reviewer for their comments. We have revised our manuscript to highlight the fact that our new *M. pinnipedii* genomes are derived from inland individuals who did not have direct contact with pinnipeds, and we discuss in detail several hypothetical models of dissemination that could explain this phenomenon, including human-to-human and animal-to-human transmissions (see Discussion). In this respect, our identification of TB in non-coastal areas bears relevance, as does our balanced interpretation. Further to this, we also discuss the persistent issue of non-MTBC mycobacterial contamination in ancient skeletal samples, and present methods on how this phenomenon can be managed. In this way the manuscript also offers an analytical model that can be followed by other groups tackling the same issue. Together, we see these as valid themes to warrant publication.

Reviewer #4 (Remarks to the Author):

The manuscript is well written and easy to follow. The analyses appear to be carefully conducted. The findings of additional ancient *M. pinnipedii* genomes that recovered from human remains in inland sites in Peru and Colombia indicates an additional mode of human-to-human TB transmission and geographic dispersal of TB disease in the pre-contact era in Southern America. The author's responses to the reviewers' comments are appropriate. The reviewer has one additional concern:

Based on current results, significant contamination was suspected on the three samples and this contaminant reads likely to arise from non-MTBC mycobacteria, which likely accounts for the unexpectedly long terminal branch-lengths for the three ancient samples in the constructed phylogenies trees (Fig. 2; Supplementary Fig. 2, 3,4). These caveats also lead to a lack of Bayesian-based molecular dating analysis in this study. However, these non-targeted reads have passed the 90% homozygosity threshold and also lead to shared variants among at least two or three ancient samples, and thus could an attempt be possible to harvest the reads harboring those variants in the long terminal branch of these three samples, and to verify whether these reads belong to non-MTBC mycobacteria or taxa (e.g. Blast)?

REPLY: We thank the reviewer for this suggestion. Accordingly, we retrieved all the reads covering homozygous SNPs unique to our three new *M. pinnipedii* genomes (82U, 281U, and 386U) as well as those shared only between 281U and 386U. We used BLAST to assess the top hits for each read and removed any read where one or more of top five hits was non-MTBC. (see Supplementary Methods, lines 364-383). It should be noted that while this process may be successful in removing reads that stem from known organisms in the database, false positive reads can persist in the dataset if the reads derive from organisms that have not been characterized, where the best match is for an MTBC member. After removing these reads, we re-did variant calling and found that this filtering removed a total of 38 SNPs, as described in the Supplementary Methods (lines 364-383), but only one of these positions was relevant for the final alignment used for phylogenetic analysis (this variant call was removed from the terminal branch of genome 82U) (Fig. 2; Supplementary Fig. 5). Phylogenetic trees shown in the main manuscript and supplementary figures are based on a complete deletion implementation of this new alignment, and there is no change in the tree topology and negligible change in branch lengths. In describing the long branch phenomenon, however, we continue to exercise caution in our choice of language, as the reads that contribute to the observed high diversity may still stem from environmental sources.

This alternative analysis is important and can provide evidence for the real branch length of these three genomes and enable the dating analysis which can add valuable insights into the common ancestor while compared to the previous finding in Bos, K. I. et al. (Nature, 2014). Such unexpected accumulation of genomic diversity, if it can be verified, can provide a different story on the estimation of the most recent common ancestor for the MTBC.

REPLY: We then attempted a dating analysis (described in detail in the Supplementary Methods, lines 385-424) and found that the estimate of the common ancestor for the *M. pinnipedii* strains is congruent with the estimate given in Bos et al. 2014. However, our estimate has wider confidence intervals, which could potentially be explained by the persistence of non-target reads in our data.

Meanwhile, such analysis may also contribute to the argument raised by reviewer #3 about the concluded ‘pinnipeds-to-humans’ transmission route. It is still less clear why the previous ancient samples published in 2014 form a sister branch with the modern *M. pinnipedii* genomes while the current three samples were located to a basal position if these samples resulted from the human(or animal)-to-human transmission after the pinnipeds-to-humans transmission.

REPLY: As discussed in our revised manuscript (lines 441-446), we believe the positioning of the ancient and modern *M. pinnipedii* strains is a result of sampling bias. To date, *M. pinnipedii* genomes have not been recovered from ancient pinniped or other non-human animal remains. The few modern *M.*

pinnipedii genomes, which all derive from pinnipeds, represent only a subset of the overall diversity of *M. pinnipedii* that existed in the past. We believe that *M. pinnipedii* genomes derived from archaeological remains from pinnipeds or other fauna could provide a more complete picture of the *M. pinnipedii* strain diversity in the past. It is possible that genomes identified in future may permit more in-depth study of the divergence that later gave rise to the 82U and the Colombian genomes we report here (281U and 386U).

Reviewers' Comments:

Reviewer #4:

Remarks to the Author:

The authors have addressed all my comments and performed additional analyses which verified the homozygous reads and reconstructed the phylogenetic trees. I have no further concerns and recommend publication.

Dear Devin Ward,

Please find our point-by-point response to the reviewer's comments below.

REVIEWERS' COMMENTS

Reviewer #4 (Remarks to the Author):

The authors have addressed all my comments and performed additional analyses which verified the homozygous reads and reconstructed the phylogenetic trees. I have no further concerns and recommend publication.

REPLY: We thank the reviewer for their comments.

Additional changes (all highlighted in yellow in the marked-up manuscript):

- The abstract was shortened as requested in the author checklist
- The Data availability section was expanded to include a description of the public datasets used
- Permit information was added to the Acknowledgements section and the funding information in this section was formatted to fit the specifications of the journal
- Additional marked changes include: minor changes associated with grammar, the removal of speech marks, the addition of company names in parentheses with relation to reagents used and the change of names used to refer to the supplementary files, figures and methods to comply with journal specifications.
- DOIs that for some reason were not displayed by EndNote were also added manually (not highlighted in yellow). All journal article references missing DOIs were checked. Journal article references that do not have DOIs listed do not have them on their respective journal/article websites.